# Hot-carrier tunable abnormal nonlinear absorption conversion in quasi-2D perovskite

Gang Wang[1], Tanghao Liu[1,2], Bingzhe Wang[1], Hao Gu[1], Qi Wei[3], Zhipeng Zhang[1], Jun He[4] ✉, Mingjie Li [3,5] ✉ & Guichuan Xing [1] ✉

Controlling the high-power laser transmittance is built on the diverse manipulation of multiple nonlinear absorption (NLA) processes in the nonlinear optical (NLO) materials. According to standard saturable absorption (SA) and reverse saturable absorption (RSA) model adapted for traditional semiconductor materials, the coexistence of SA and RSA will result in SA induced transparency at low laser intensity, yet switch to RSA with pump fluence increasing. Here, we observed, in contrast, an unusual RSA to SA conversion in quasi-two-dimensional (2D) perovskite film with a low threshold around 2.6 GW cm⁻². With ultrafast transient absorption (TA) spectra measurement, such abnormal NLA is attributed to the competition between excitonic absorption enhancement and non-thermalized carrier induced bleaching. TA singularity from non-thermalized "Fermi Sea" is observed in quasi-2D perovskite film, indicating an ultrafast carrier thermalization within 100 fs. Moreover, the comparative study between the 2D and 3D perovskites uncovers the crucial role of hot-carrier effect to tune the NLA response. The ultrafast carrier cooling of quasi-2D perovskite is pointed out as an important factor to realize such abnormal NLA conversion process. These results provide fresh insights into the NLA mechanisms in low-dimensional perovskites, which may pave a promising way to diversify the NLO material applications.

High-power laser controled by nonlinear absorption (NLA) through nonlinear optical (NLO) materials is of significant importance in laser development and applications. Generally, NLA can be categorized into saturable absorption (SA) and reverse-saturable absorption (RSA)[1]. SA, originating from the Pauli-blocking effect, provides the increased transmittance under high optical excitation and is crucial for ultrafast laser pulse generation[2] and all-optical switching[3]. In contrast, RSA, resulting from the multiphoton absorption (MPA), excited state absorption (ESA) or free carrier absorption can lead to

the transmittance decreasing, which is useful for laser protection[4] and microimaging[5]. Under the particular circumstances[6–9], coexistence of SA and RSA will result in the conversion between them, which is significant to inspire new techniques in all-optical devices such as dissipative soliton generation[10] and optical logic gate[11]. Thereinto, the typical overlapping between standard SA and RSA will induce a switching from SA to RSA with incident light intensity increasing (Fig. 1a). Such phenomenon has been observed in various three-dimensional (3D)[6] and two-dimensional (2D)[7] materials, where

[1]Joint Key Laboratory of the Ministry of Education, Institute of Applied Physics and Materials Engineering, University of Macau, Avenida da Universidade, Taipa, Macao SAR 999078, China. [2]Department of Physics, Hong Kong Baptist University, 224 Waterloo Road, Kowloon, Hong Kong SAR 999077, China. [3]Department of Applied Physics, The Hong Kong Polytechnic University, Hong Kong, China. [4]Hunan Key Laboratory of Nanophotonics and Devices, Central South University, 932 South Lushan Road, Changsha 410083, China. [5]Photonics Research Institute, The Hong Kong Polytechnic University, Hung Hom, Kowloon, Hong Kong. ✉e-mail: junhe@csu.edu.cn; ming-jie.li@polyu.edu.hk; gcxing@um.edu.mo

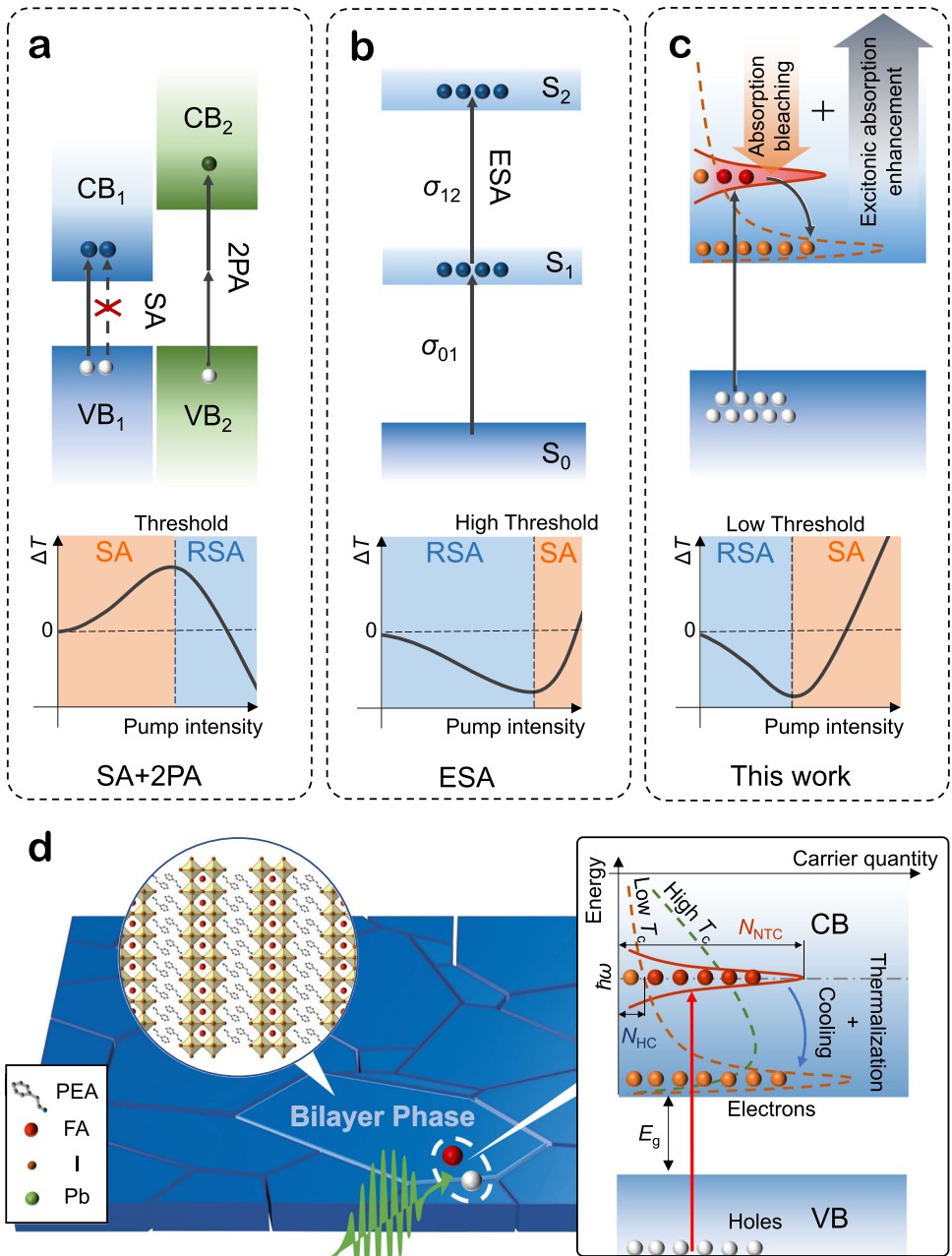

**Fig. 1 | Nonlinear absorption switching mechanism. a** Schematics of NLA switching in conventional inorganic semiconductors (top) and the corresponding NLA evolution as a function of pump intensity (bottom), according to standard SA and RSA, where the RSA can stem from two-photon absorption (2PA). CB and VB denote conduction band and valence band respectively. $\Delta T$ denotes the NLA induced transmittance difference. **b** NLA of organic molecules based on excited state absorption (ESA) leading to an inverse transition from RSA to SA. $S_{0-2}$ represent the separated energy level. $\sigma$ is the absorption cross-section. **c** NLA process of quasi-2D (PEA)$_2$FAPb$_2$I$_7$ perovskite film in our work, where the competition between non-thermalized carrier induced bleaching and many-body effect induced excitonic absorption enhancement results in a conversion from RSA to SA. **d** Schematic of the initial carrier evolution process in deposited quasi-2D perovskite film under femtosecond laser pulse excitation, where the blue blocks represent the bilayer perovskite grains. The left inserted illustration demonstrates the 2D lattice structure, and the right inset illustrates the initially ultrafast intra-band relaxation of photon generated carriers. Red arrow indicates the transition from VB to CB. $E_g$ is the bandgap. Solid and dash lines represent the non-thermalized carrier and hot-carrier distribution, respectively. $N_{NTC}$ and $N_{HC}$ denote the non-thermalized carrier and hot carrier qualities at the resonance with incident photon energy ($\hbar\omega$). $T_C$ is hot carrier temperature reflecting the carrier distribution after thermalization. Compared with the high $T_C$ distribution (green), low $T_C$ distribution (orange) will reduce the value of $N_{HC}$ and weaken the bleaching induced by band filling.

the SA stems from the ground state bleaching, and the RSA is primarily attributed to MPA. Nevertheless, the opposite power-dependent conversion from RSA to SA is merely reported in few kinds of organic molecules (Fig. 1b)[8,9], in which the RSA stems from strong ESA and the NLA transition to SA need to pump considerable quality of electrons to higher excited states by the incident photons.

Therefore, for ultrafast NLA devices working at femtosecond regime, a high NLA conversion threshold up to hundreds GW cm$^{-2}$ (Supplementary Table 1) is normally required[9], which severely hampers its practical applications e.g. in low-power optical switching and modulation[3,12]. The detailed NLA conversion mechanisms are exhibited in Supplementary Note 1.

During the past decade, owning to the remarkable optoelectronic properties (e.g. long carrier lifetime[13], high photon capture efficiency[14], strong defect tolerance[15]), hybrid perovskites have achieved significant progress in various fields spanning from solar cell[16], light emission devices[17], semiconductor laser[18] to photodetector[19]. Meanwhile, the strong light-matter interaction also endows perovskites remarkable NLO properties[20]. By introducing the large organic molecules, the quasi-2D perovskite with a structure formula of $L_2S_{n-1}M_nX_{3n+1}$ is synthesized, where the giant organic cation barriers (L) confine the carriers in the inorganic metal halide octahedrons ($MX_6$) layer to form a natural quantum-well (QW) structure[21]. Such low-dimensional QWs structures with strong quantum and dielectric confinement[22] have been demonstrated to result in enhanced NLO responses[23–25]. However, the underlying mechanism and tailoring of NLA process are not well understood and explored.

In this work, we observe an unexpectedly power-dependent NLA conversion from RSA to SA in $(PEA)_2FAPb_2I_7$ quasi-2D perovskite film with a conversion threshold of $2.6 \pm 0.2$ GW cm$^{-2}$, which is in contrast to the normal phenomenon observed in the 3D counterpart. Using the transient absorption (TA) spectroscopy and theoretical simulation, we elucidate that such anomalous NLA transition process is correlated with the ultrafast non-thermalized carrier induced bleaching and many-body effect induced above band-edge absorption enhancement (Fig. 1c). The ultrafast carrier thermalization below 100 fs is clearly observed from the singularities of transient absorption. Moreover, we also reveal that the hot carrier effect is a nontrivial factor to tune the NLA in hybrid perovskite materials (Fig. 1d). The accelerated hot-carrier cooling in the quasi-2D perovskite QWs enhances the initial RSA and results in a NLA conversion from RSA to SA. By contrast, the high-temperature hot carrier distribution in the 3D counterpart leads to a broadband SA response. Our results uncover an interesting mechanism of NLA conversion process of low-dimensional semiconductor materials that may inspire a promising way approaching high-speed and low-power NLO devices.

## Results

### Abnormal NLA behaviors in quasi-2D perovskites

Our quasi−2D perovskite film is deposited using an antisolvent-assisted solution process with a 3D $FA_{0.9}MA_{0.1}PbI_3$ film as a counterpart. The preparation details are given in methods and characterization results from scanning electron microscope (SEM), atomic force microscope (AFM), X-ray diffraction (XRD) and thickness (Fig. 2a–d) indicate excellent crystalline quality with film roughness $R_q < 10$ nm to avoid light scattering. The inset of Fig. 1d have illustrated the schematics of the deposited quasi-2D perovskite film with dominated natural QWs grains, which is reflected from a sharp excitonic absorption peak at 570 nm (2.18 eV) corresponding to the ($n = 2$) $(PEA)_2$-$FAPb_2I_7$ bilayer phase (Fig. 2e)[26,27]. The NLO properties of samples are characterized by the femtosecond open-aperture Z-scan method. The NLA response of quasi-2D perovskite film was collected in a wide range from 540 nm to 590 nm (Supplementary Fig. 1). As shown in Fig. 2f, an intriguing phenomenon was observed on the blue side above 1 S exciton in the quasi-2D perovskite film. There is an apparent RSA at low pump intensity, and with increasing laser intensity, the RSA converts to SA with a threshold of $2.6 \pm 0.2$ GW cm$^{-2}$. Such NLA conversion from RSA to SA in quasi-2D perovskite is contradictory to the traditional NLA transition theory developed for bulk semiconductors, which results in an opposite conversion from SA to RSA with pump fluence increasing. Using a standard NLA and SA model (Supplementary Note 2), the NLO parameters such as NLA coefficient, saturable intensity and modulation depth were extracted (Supplementary Table 2). The obtained RSA coefficient at 540 nm is 12.75 cm MW$^{-1}$ which is at least two-orders of magnificent larger than the previous one reported two-photon absorption (2PA) of conventional semiconductors[28,29] and 3D bulk perovskites[30,31]. By contrast, in the 3D bulk counterpart, when the

incident photon energy surpasses the band gap, the sample only delivers a broadband SA (Fig. 2g and Supplementary Fig. 2) similar to that of previously reported inorganic 3D bulk semiconductors[32] and emerging 2D layer materials[33]. Such quite intriguing NLA response of quasi-2D perovskite film appeals to our considerable attention, which may uncover new physical mechanism tuning the NLO of low-dimensional systems.

### TA features on the explanation of NLA behaviors

To uncover the origin of such abnormal NLA behaviors in quasi−2D perovskite film, a systematic broadband ultrafast TA spectroscopy is employed on samples. Figure 3a shows the pseudocolour plot of transient absorption difference ($\Delta A$) as a function of probe light wavelength and delay time under the excitation of a 400 nm femtosecond laser pulse. The pump intensity of 0.64 mJ cm$^{-2}$ (peak power of 6.4 GW cm$^{-2}$) corresponds to an excited carrier density of $1.45 \times 10^{20}$ cm$^{-3}$ (Supplementary Note 3). Negative and positive $\Delta A$ represent the photon-induced bleaching (PIB) and absorption (PIA), respectively. According to the TA profile near band-edge (Fig. 3b), a broadband PIA shoulder (PIA$_1$) locates at the blue side of the band-edge PIB at 570 nm (PIB$_1$), which is consistent well with the NLA coefficient distribution (Fig. 3c, red region) extracted from Z-scan measurements. This result indicates that the above bandgap RSA observed at Z-scan measurement has the same origination as the PIA$_1$ feature from TA spectrum. The dynamics TA signal could thus provide critical information for understanding the abnormal NLA behaviors, and the origins of TA peaks should be addressed first. Herein, PIB$_1$ peak at 570 nm corresponds to the state-filling induced bleaching of 1 S exciton of bilayer perovskite phase. A clear understanding of the origin PIA signal of perovskite is not trivial. In the pure organic molecules, the PIA is normally attributed to ESA and requires larger absorption cross-section of excited state than that of ground state. Whereas, for quasi-2D perovskite, the electronic band structure and optical transition are mainly determined by the lead halide perovskite layer instead of the large organic spacers acting as confinement barriers[34]. The transition from lowest excited states to adjacent higher energy level is expected to be fobidden[35]. Meanwhile, the direct bandgap makes lead halide perovskite have large absorption cross-section of the ground state. Hence, the contribution of ESA should be excluded. To further clarify the underlying mechanism of PIA, we performed a pump wavelength dependent measurement from 400 nm to 580 nm (Supplementary Fig. 3). With reducing the pump photon energy, the amplitude of PIA$_2$ peak below the band edge decreases rapidly, which thus implies that PIA$_2$ could originate from the biexciton interaction between the hot exciton generated by the pump photon and the low-energy exciton generated by the probe photon, leading to an absorption red-shift proportional to the number of injected excitons[36]. By contrast, the broad-band PIA$_1$ suggests persistent intensity, which indicates a completely different mechanism. Note that the above band-edge PIA$_1$, also widely observed in 3D bulk perovskites, was previously ascribed to the band-gap renormalization (BGR)[37]. Specifically, BGR stems from the exchange and correlation effect among free carriers, which will lead to a bandgap shrinking by $\Delta E_g \propto n^\alpha$, where the value of $\alpha$ is 1/3 for perovskites[38,39]. However, our pump power dependent TA measurements (Fig. 3d) reveal that PIA$_1$ obeys a saturation behavior (solid blue line) similar to that of PIB$_1$ (Supplementary Fig. 4) instead of the BGR model (dash gray line). This result indicates BGR effect may be not the main root of PIA$_1$ in quasi-2D perovskite film. According to recent studies, for such QWs semiconductor material with considerable exciton binding energy, a more rational explanation of the above band-edge PIA$_1$ is excitonic absorption enhancement induced by many-body effect[40,41]. The many-body interaction will be modulated by the injected carrier distribution, which could explain why PIA$_1$ suggests a saturate behavior as pump fluence rises.

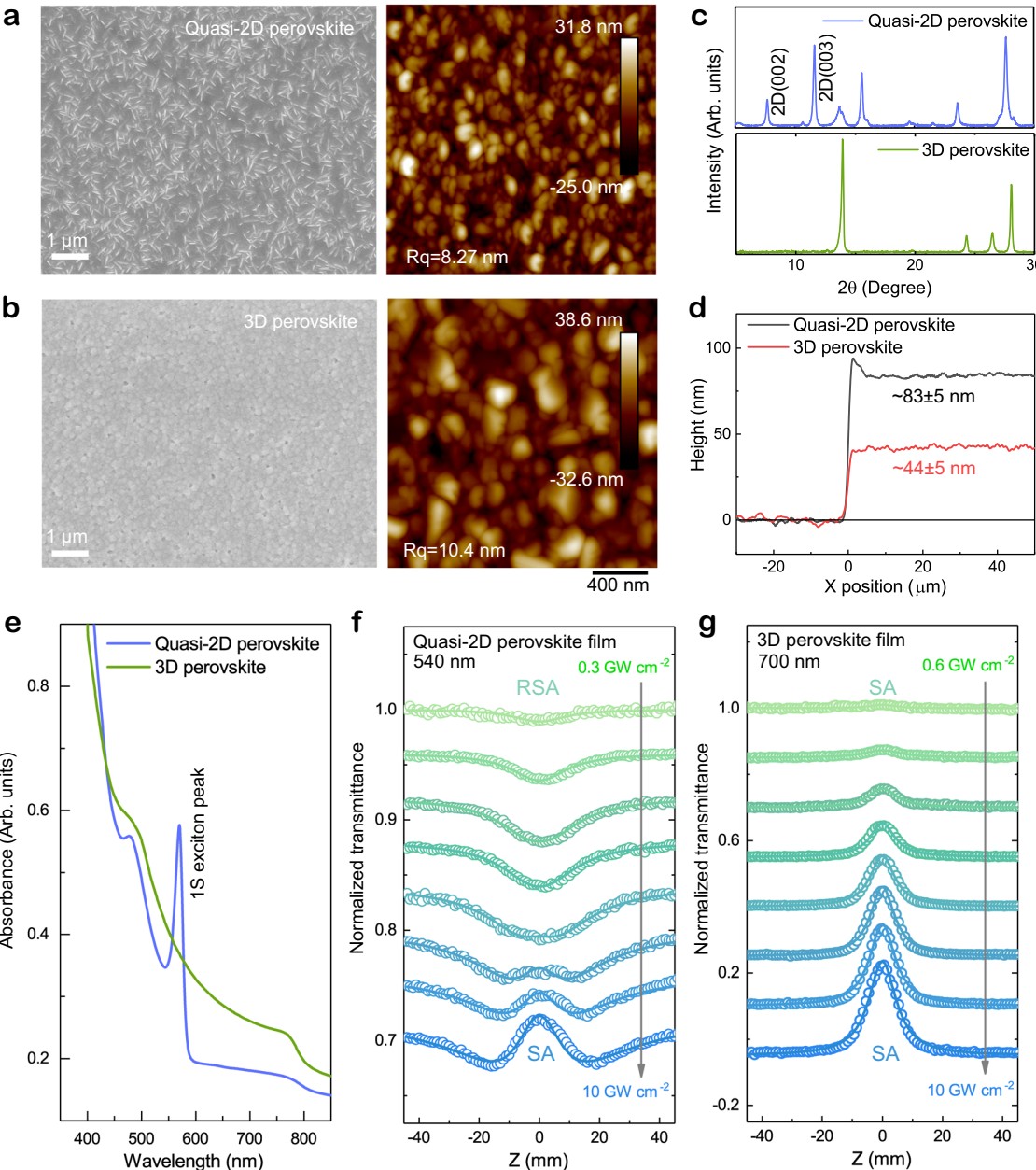

**Fig. 2 | Physical properties and nonlinear optical characteristics. a, b** Scanning electron microscope (SEM) and atomic force microscope (AFM) images of bilayer quasi-2D (**a**) and 3D (**b**) perovskite thin films. **c, d** are the corresponding X-ray diffraction (XRD) patterns and film step profiles. **e** Linear ultraviolet-vis absorbance (Abs.) spectra of the 3D perovskite and quasi-2D perovskite film with a strong 1 S exciton absorption peak at 570 nm. **f** Z-scan profile (hollow circles) evolutions of bilayer quasi-2D perovskite film collected at the blue side (540 nm) of 1 S exciton peak demonstrating a conversion from RSA to SA with excitation intensity rising. The solid lines are fitting curves using the abnormal NLA conversion model (Eq. 2) we developed. **g** Corresponding Z-scan results (hollow circles) of 3D counterpart at 700 nm above band-edge with a pure SA response fitting well with the standard SA model (solid lines) according to Eq. 4 in supplementary information.

Next, to investigate the NLA variation at the laser wavelength approaching the band-edge, the pump laser wavelength is changed from 400 nm to 540 nm in TA measurement. The pump fluence is 0.76 mJ cm$^{-2}$ (peak power of 7.6 GW cm$^{-2}$) corresponding to a photon-injecting carrier density of $1.33 \times 10^{20}$ cm$^{-3}$. An unexpectedly additional TA feature (Fig. 3e, red dash frame) was observed. As shown in Fig. 3f, a new derivative-like signal (in gray dashed frame, a PIB valley centered at 543 nm accompanied by a blue-side weaker PIA peak centered at 535 nm) overlapped with basic PIA$_1$ shoulder, which persists at least up to the end of the pump pulse (Methods). Such additional TA feature will move along with the pump laser wavelength (Supplementary Fig. 5) and appears more obviously at higher pump intensity

(Supplementary Fig. 6). From the power-dependent TA kinetics probed at 540 nm (Fig. 3g), it is found that an ultrafast bleaching (green region) was created within the first 300 fs. With the pump fluence rising, such transient PIB becomes increasingly stronger leading to $\Delta A$ at 150 fs changing from positive to negative (insert of Fig. 3g), which is in line with the conversion from RSA to SA observed in aforesaid NLA measurement. Such intriguing phenomenon is a characteristic signal of the presence of non-thermalized carriers, which is known as absorption edge singularity reported in high excited conventional semiconductor GaAs[42,43]. Specifically, the photoexcited non-thermalized carriers will quickly develop to a quasi-equilibrium thermal distribution described with a temperature $T_C$ via carrier-carrier

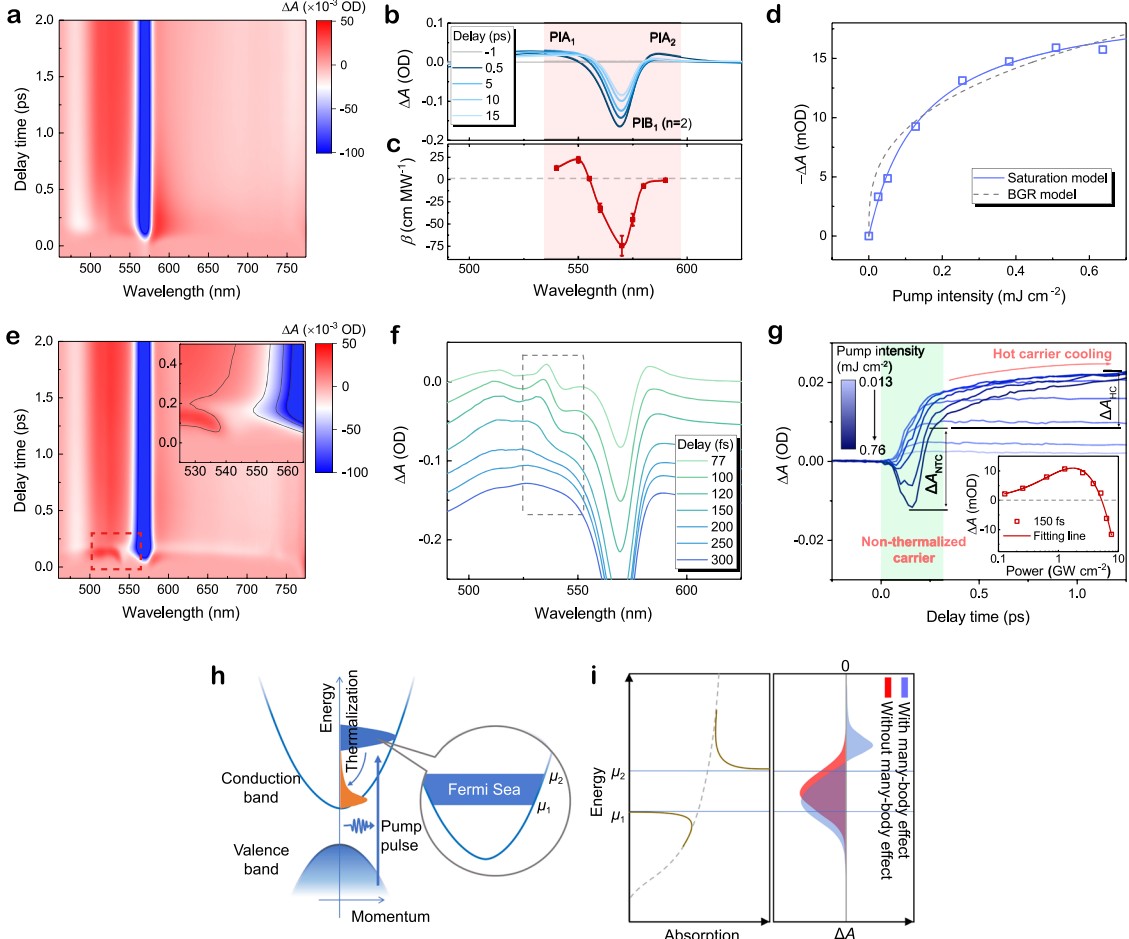

**Fig. 3 | Ultrafast TA measurement of quasi-2D perovskite film. a** Representative pseudocolour TA spectrum and **b** Time-dependent photo-induced changes in absorption ($\Delta A$) which is pumped by 400 nm femtosecond laser with excitation fluence around 0.64 mJ cm$^{-2}$. **c** NLA coefficients extracted from Z-scan measurements. Error bars represent the fitting uncertainties with standard NLA model. **d** Experimental $\Delta A$ data (blue hollow dots) of PIA$_1$ peak (530 nm) at 200 fs delay as a function of pump intensity fitted by the band-gap renormalization (BGR) model (gray dash line) of $\Delta A = A \times I^{1/3}$ and standard saturation model (blue solid line) of $\Delta A = B \cdot I/(1 + I/I_S)$, where $B$ are constant, $I_S$ is saturable intensity with a value of 0.16 mJ cm$^{-2}$ (1.62 GW cm$^{-2}$). **e** Pseudocolour representation of TA spectrum pumped by 540 nm femtosecond laser with an excitation intensity of 0.76 mJ cm$^{-2}$. Inset: enlarged TA spectrum of the part labeled by the red frame. **f** $\Delta A$ spectra

evolution within the first 300 fs and **g** pump fluence dependent kinetics curves probed at 540 nm extracted from **e**. Green region highlights the non-thermalization carrier kinetics within the first 300 fs. Red arrow in **g** outlines the subsequent hot-carrier cooling process after thermalization. The non-thermalized carriers and hot carriers induced absorbance difference is labeled by $\Delta A_{NTC}$ and $\Delta A_{HC}$. Inset: power dependent $\Delta A$ (hollow dots) at 150 fs delay fitted with Eq. 2 (red solid line). **h** The schematic shows the initial intra-band carrier relaxation process. **i** Sketched absorption spectrum based on nonequilibrium Fermi-sea. The left panel denotes the absorption singularities appear near the upper and lower boundary. The right panel demonstrates $\Delta A$ spectra with/without many-body interaction in the presence of a gaussian type non-thermalized carrier distribution.

scattering, which is known as thermalization. However, due to the monochromaticity of pump laser, at the initial stage, the injected carriers still distribute in a narrow energy range between $\mu_1$ and $\mu_2$, which can be modeled by a "nonequilibrium Fermi sea" (Fig. 3h). Owing to the many-body interactions between Fermi sea and the deep hole created by probe photon absorption, the singularities appearing on both sides of the photon-induced transparent region can be simply described by a power law[42]:

$$\alpha(\omega) \propto 1/(\hbar\omega - \mu_i)^{\eta_i} \qquad (1)$$

where $i = 1, 2$. At the vicinity of the upper limit $\mu_2$, the value of exponent $\eta_2$ is positive, which leads to an enhanced absorption, whereas at the lower threshold $\mu_1$, $\eta_1$ is negative, the absorption is reduced (Fig. 3i, left panel). Therefore, the practically observed $\Delta A$ ($= L \Delta \alpha$, where $L$ is the thickness of sample) spectrum deviates from dense electron-hole plasma induced hole burning in the absorption spectrum (Fig. 3i, right panel, red region) and result in a derivative-like signal (Fig. 3i, right

panel, blue region). Hence, the NLA response at 540 nm is governed by the competition between the broadband excitonic absorption enhancement (PIA$_1$) and the non-thermalized carrier induced bleaching part, which can be described by a phenomenological model:

$$\Delta \alpha = \frac{\beta I}{1 + I/I_{s1}} - \frac{a_0 I/I_{s2}}{1 + I/I_{s2}} \qquad (2)$$

Where $\beta$ is used to evaluate the excitonic absorption enhancement induced RSA which is limited by saturable intensity $I_{s1}$, the second term denotes the non-thermalized carrier induced bleaching (giving SA), $a_0$ is the linear absorption coefficient at 540 nm. As illustrated in inset of Fig. 3g (red line), the measured power-dependent $\Delta A$ conversion process can be well fitted with this model (all fitting parameters are summarized in Supplementary Table 3). The obtained value of $I_{s1}$ is fitted to be 2.06 GW cm$^{-2}$. By contrast, the value of $I_{s2}$ reaches up to 40.64 GW cm$^{-2}$. By fitting the TA kinetics using three-exponential model convoluted with gaussian type response function

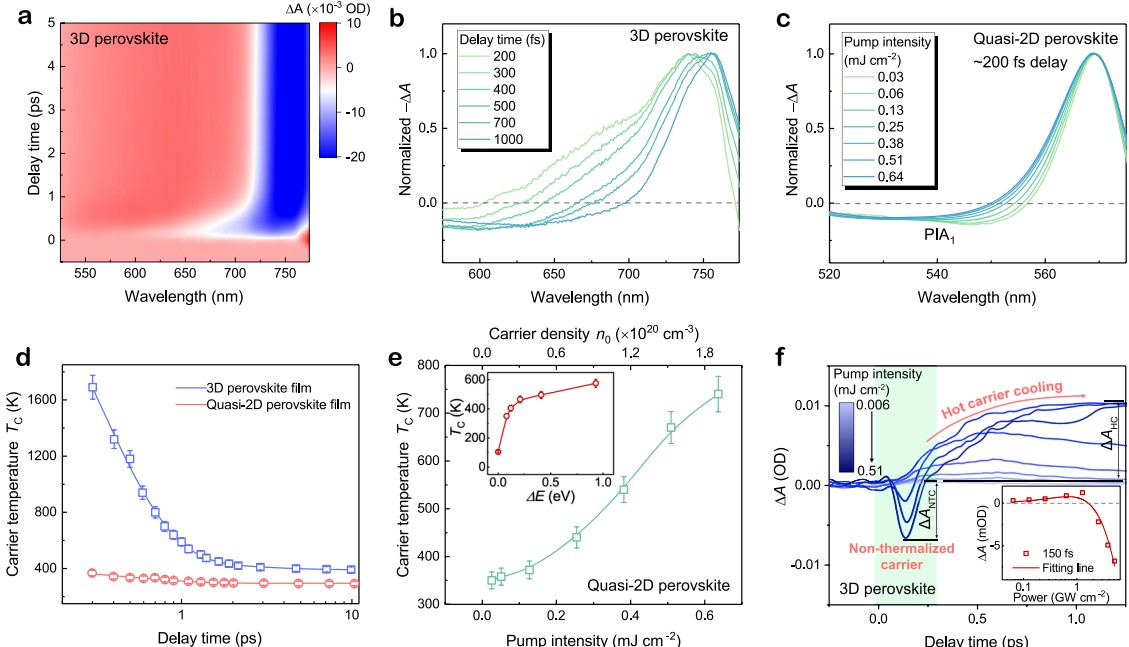

**Fig. 4 | Comparison of hot-carrier effects. a** Pseudocolour representation and **b** Normalized TA spectra at different delay times of 3D bulk perovskite film. The pump laser pulse is 480 nm (2.6 eV) with excitation fluence around 0.13 mJ cm⁻². **c** Normalized TA spectra of quasi-2D perovskite film at 200 fs delay under different excitation intensities. The pump laser wavelength is 400 nm (3.1 eV). Herein, the energy of the photon $\hbar\omega$ used to excite 3D and quasi-2D perovskites is 2.6 eV and 3.1 eV, respectively, to ensure an equal excess energy (0.9 eV) of initially injected carriers. **d** Time-dependent hot carrier temperatures of 3D and quasi-2D per-ovskites, respectively. The pump fluence is maintained at 0.13 mJ cm⁻². **e** Initial hot-

carrier temperature of quasi-2D perovskite film as a function of pump power (pump photon energy of 2.6 eV) and excess energy (inset, the pump fluence is kept at 0.4 mJ cm⁻²). Error bars in **d**, **e** represent the uncertainties in the fitting of carrier temperature. **f** TA kinetics under different excitation fluence pumped with 540 nm femtosecond laser pulse. Green region highlights the non-thermalization carrier kinetics within the first 300 fs. Red arrow outlines the subsequent hot-carrier cooling process after thermalization. The non-thermalized carriers and hot carriers induced absorbance difference is labeled by $\Delta A_{NTC}$ and $\Delta A_{HC}$. Inset: power-dependent $\Delta A$ (hollow dots) at 150 fs delay fitted with Eq. 2 (red solid line).

(Supplementary Fig. 7), the carrier thermalization time can be extracted from $48 \pm 5$ fs to $66 \pm 5$ fs depending on the pump fluence, which is consistent with the value obtained using two-dimensional electronic spectroscopy[44]. Such ultrafast thermalization process makes the carriers difficult to accumulate at their initial energy position with resonance of 540 nm photon and thus lead to a higher value of saturable intensity $I_{S2}$ to generate effective SA. As shown in Fig. 2f and Supplementary Fig. 2 (540 nm-555 nm), utilizing Eq. 2, the Z-scan results were reproduced successfully. The solid fitting curves coincides well with the experimental data, which further confirms the reliability of our NLA conversion model.

It is also worth noting that, when the laser power is far below the saturation of $I_{s2}$, Eq. 2 can be approximated as:

$$\triangle\alpha = \frac{\beta I}{1 + I/I_{s1}} - (\alpha_0/I_{s2}) \cdot I \qquad (3)$$

At the low pump fluence, as $\beta$ is larger than $\alpha_0/I_{s2}$ (see Supplementary Table 3), the material only suggests RSA. However, with increasing the laser intensity, excitonic absorption enhancement induced RSA is suppressed by the saturation effect, and the non-thermalized carrier induced SA exceeds RSA, which results in a NLA conversion from RSA to SA. Therefore, the large divergency between $I_{s1}$ and $I_{s2}$ can be considered as the key factor to realize the abnormal NLA conversion behavior of quasi-2D perovskite. Such NLA switching mechanism is completely different from that of previously reported conventional bulk semiconductors[45] and isolated organic molecules[9]. Compared with organic molecules, the NLA of quasi-2D perovskite only involves two energy band (CB and VB), which may lead to a relatively lower conversion threshold below 10 GW cm⁻².

## Influence from hot-carrier effects

To further elucidate the underlying mechanism of different NLA behaviors between quasi-2D and 3D perovskite, similar TA analysis is also performed on 3D counterpart. As shown in Fig. 4a and b, within the hundreds of femtoseconds after photoexcitation, the 3D per-ovskite film illustrated a broadband PIB even overwhelming the posi-tive PIA₁ shoulder, which indicates a high-energy tail arising from hot-carrier occupations. By contrast, for the quasi-2D perovskite as illu-strated in Fig. 4c, the hot-carrier induced bleaching is much weaker even under high-power laser excitation. Herein, the carrier tempera-ture $T_C$ is extracted by fitting the transient TA spectra (Supplementary Note 4, Supplementary Fig. 8 and 9), in which the initial excitation intensity and excess energies are controlled to be close in both sam-ples for a fair comparison. As compared with the 3D counterpart, the quais-2D perovskite exhibits a much lower $T_C$ (Fig. 4d), and the initial $T_C$ at the higher pump intensity and different excess energies can only approach to ~800 K (Fig. 4e). These results indicate that, during the ultrafast thermalization stage in quasi-2D perovskite, carriers have already lost most of extra energy and undergo a sufficient cooling down to the band edge. Such ultrafast carrier cooling process is also reflected by the rapid building-up of band-edge PIB of quasi-2D per-ovskite in contrast to that of 3D counterpart (Supplementary Fig. 10). Similar phenomenon was also observed in monolayer 2D layer perovskite[46,47], where the sub-picosecond intraband hot-carrier relaxation is attributed to the stronger nonadiabatic couplings among high-energy states and electron-phonon coupling with the organic spacer barriers. Besides, for quasi-2D perovskite QWs, the reduced coulomb screening due to dielectric confinement effect will also accelerate hot carrier cooling[48]. From the TA kinetics curves shown in Fig. 4f, we found that, compared with 2D RP perovskite (Fig. 3g), 3D perovskite demonstrates a more obvious hot-carrier

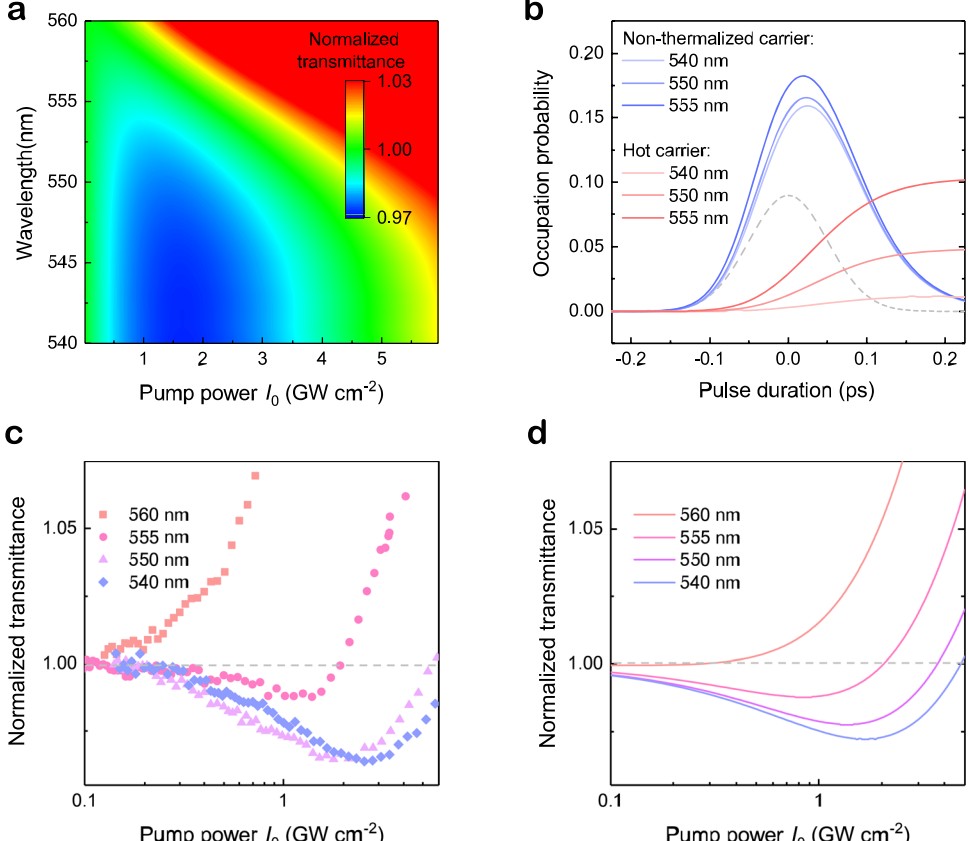

**Fig. 5 | Mathematical simulation of carrier kinetics and nonlinear absorption.** **a** Calculated pseudocolour representation of the NLA mapping as a function of pump power and laser wavelength. **b** Occupation possibility evolution of non-equilibrium carrier and hot carrier under a pump fluence of 0.6 mJ cm$^{-2}$ (corresponding to a peak power of 6 GW cm$^{-2}$). Gray dash line denotes the excitation laser pulse. **c** Experimental and **d** simulated laser power-dependent normalized transmittance at different wavelengths.

cooling process with the hot-carrier vs non-thermalized carrier TA amplitude ratio ($\Delta A_{HC}/\Delta A_{NTC}$) increasing from 0.65 (quasi-2D perovskite) to 1.44 (3D perovskite). As shown in the inset of Fig. 4f, at 150 fs delay time, only a weak PIA (<0.001 OD) can be observed under low pump intensity. These results mean that, in 3D perovskite, the high-temperature hot carrier will induce additional bleaching covering the PIA$_1$, which results in a broadband SA response. The hot-carrier effect is confirmed as an important factor controlling the band-edge NLA response of perovskite materials. For the bilayer quasi-2D perovskite QWs, the ultrafast hot-carrier cooling prevents the hot-carrier induced bleaching, which allows the excitonic absorption enhancement induced RSA to be observable at low excitation intensity. Furthermore, the strong excitonic interaction owing to quantum and dielectric confinement will strengthen the many-body interaction induced above band-edge PIA shoulder[49], which also makes the RSA more difficult to be covered by the hot carriers induced bleaching.

## Discussion

Above TA results reveal that the NLA response of quasi-2D perovskite materials is determined by the complex interplay among band-edge absorption enhancement, non-thermalized hot-carrier induced singularities and hot-carrier cooling effect. Lastly, we develop a simple model taking all these factors into consideration to quantificationally describe the NLA process at different laser wavelengths and intensities. Based on the fermion nature and Fermi's golden rule[50], the non-thermalized and thermalized carriers are considered as in two separated energy levels. The schematic of initial intra-band thermalization and cooling processes is shown in the right illustration of Fig. 1d. In

quasi-2D perovskite film, the photon injected transient non-thermalized carriers (solid red line) will decay to a hot-carrier distribution (orange dash line) by ultrafast thermalization and simultaneously rapid carrier cooling. The carriers occupying the states at $\hbar\omega$ can be divided into two parts belonging to non-thermalized carriers ($N_{NTC}(\hbar\omega)$) and hot carriers ($N_{HC}(\hbar\omega)$), respectively. The transient absorption coefficient of sample under laser illumination can be expressed as:

$$\alpha(I(t),\hbar\omega) = \left[1 - 2\left(f_{NTC}(\hbar\omega) + f_{HC}(\hbar\omega)\right)\right] \times \alpha_0(\hbar\omega) + \frac{C \times I(t)}{1 + I(t)/I_S} \quad (4)$$

where the bleaching of the linear absorbance $\alpha_0(\hbar\omega)$ (first term) is determined by both non-thermalized carrier ($f_{NTC}(\hbar\omega) = \frac{N_{NTC}(\hbar\omega)}{Dos}$) and hot carrier ($f_{HC} = \frac{N_{HC}(\hbar\omega)}{Dos}$) occupation. Dos denotes the density of states in conduction band. The non-thermalized carriers distribute in a narrow energy range as a Lorentzian function and the hot carriers obeying Fermi-Dirac distribution $f_{HC}(\varepsilon) = 1/\left[1 + \exp\left(\frac{\varepsilon - E_f}{K_B T_C}\right)\right]$. Here, to reduce the simulation complexity, the initial hot-carrier temperature after thermalization is fixed at 350 K corresponding to the temperature under low pump fluence extracted from Fig. 4e. The second term from RSA is described using an aforesaid phenomenological saturation model, in which $C$ is the constant to evaluate the RSA intensity, $I_S$ is the saturable intensity. The detailed simulation process is demonstrated in Supplementary material Note 5, where carrier dynamics is solved by a numerical method. All the parameters used are collected in supplementary Table 4. The calculated NLA spectra are plotted in Fig. 5a which reveal a remarkable RSA-SA conversion with pump power at the

high energy regime (<560 nm). Figure 5b shows the time-dependent occupation probability evolution of non-thermalized carrier and hot carrier at different excitation wavelengths. With a low carrier temperature, the hot-carrier occupation probability is negligible at high energy regime such as 540 nm. Therefore, the NLA is determined by the competition between broadband excitonic absorption enhancement (RSA) and the non-thermalized carrier induced bleaching (SA), which can be described according to Eq. 2. As mentioned above, the RSA dominates NLA response at low pump fluence. With laser power increasing, the RSA reaches saturation and the non-thermalized carrier induced SA rises rapidly resulting in NLA conversion from RSA to SA. Whereas, due to most of injected hot carriers accumulating close to the band edge, at the longer laser wavelengths (>555 nm), the bleaching from band filling of hot-carrier becomes stronger and offset the RSA part, which thus accounts for the observed increased SA. Therefore, the pump-power dependent transmittances from Z-scan measurement (Fig. 5c) can be represented well using our theoretical model (Fig. 5d) at 540 nm-555 nm. Notably, at 560 nm, the measured NLA curve in Fig. 5c illustrate strong SA even at low pump fluence, which deviates slightly from our simulated results in Fig. 5d. This difference can be attributed to the contribution of exciton effect. At the regime adjacent to 1 S exciton, the optical properties would be governed by the exciton states. Therefore, the reduction of exciton oscillator strength due to screening should also be taken into account and it can lead to enhancement of actual SA as compared with the calculations[51]. These results thus indicate that, by utilizing the materials with different hot-carrier dynamics, the NLA can be modulated flexibly. If the carrier temperature $T_C$ is higher due to slow hot-carrier cooling, the hot carrier occupancy will increase (green dash line in Fig. 1a), which can enhance the SA and weaken the RSA substantially (Supplementary Fig. 11). By contrast, if the carrier-carrier scattering is accelerated, the non-thermalized carriers induced SA will be suppressed which makes the RSA more obvious at low laser intensity. Thus far, we have clarified the mechanisms of abnormal NLA process in the quasi-2D perovskite clearly.

To verify the universality of the phenomenon we observed, the NLA behavior of other 2D perovskite materials are also studied. We synthesize the $(PEA)_2FAPb_2Br_7$ ($n = 2$) quasi-2D perovskite (replacing the iodine (I) element with bromine (Br) element) and monolayer $(PEA)_2PbI_4$ ($n = 1$) 2D perovskite with similar spin-coating method. As shown in supplementary Fig. 12, the obtained $(PEA)_2FAPb_2Br_7$ and $(PEA)_2PbI_4$ film suggest remarkable 1 S exciton peak at 435 nm and 515 nm. For $(PEA)_2FAPb_2Br_7$ perovskite, the Z-scan curves (supplementary Fig. 13) suggest the similar RSA-SA conversion at the blue side of the band-edge (410 nm-420 nm), which is in line with that we observed in $(PEA)_2FAPb_2I_7$ film. From the TA spectrum (supplementary Fig. 14), we can also observe the ultrafast absorption singularity signal. Moreover, we also investigate the NLA properties of $(PEA)_2PbI_4$ ($n = 1$) 2D perovskite film (supplementary Fig. 15). By contrast, as illustrated in supplementary Fig. 15, the sample only suggest RSA at the blue side of the exciton peak (515 nm) until high pump intensity up to 10 GW cm$^{-2}$. By retrospecting the linear absorption spectra of different 2D perovskite (supplementary Fig. 12), we find the absorption shoulder from continuous state is much lower for $(PEA)_2PbI_4$ ($n = 1$) perovskite, which indicates a lower continuous state density. Low continuous state density will suppress the non-thermalized carrier induced bleaching and make the RSA to SA switching difficult to occur. This deduction is also confirmed by the TA spectrum (supplementary Fig. 16), where the transient absorption singularity signal owing to the non-thermalized carrier occupation become almost absent. These results indicate that the RSA-SA conversion is not a universal property for every 2D materials. Therefore, to realize RSA-SA conversion, besides the ultrafast carrier cooling, appropriate continuous state density is needed. By modulating the component and dimensionality, the NLA response can be tuned flexibly for 2D perovskite materials. We foresee that the

abnormal RSA-SA conversion observed in quasi−2D perovskite can complement with the typical SA-RSA conversion observed in conversional semiconductors. Compared the typical SA-RSA conversion, the RSA-SA conversion is more appropriate for the application in laser pulse compressing and eliminating the low-intensity wing signal[52]. Moreover, such abnormal NLO response is also expected to inspire valuable strategy to design all-optical devices *e.g.* all-optical switching or logic-gate.

In conclusion, we observe a rare NLA conversion from RSA to SA in quasi-2D perovskite film under above band-edge excitation with a low conversion threshold of $2.6 \pm 0.2$ GW cm$^{-2}$. Using TA spectroscopy measurement, the underlying NLA mechanisms were uncovered unambiguously. Such abnormal NLA switching phenomenon is found to be due to the competition between the many-body effect induced excitonic absorption enhancement and non-thermalized carriers induced bleaching, which is modulated by the initial hot carrier distribution. The ultrafast hot carrier cooling process of quasi-2D perovskite is an important factor leading to the observed abnormal RSA-SA conversion. Furthermore, theoretical analysis of the ultrafast intraband carrier dynamics reveals that, by tuning the carrier thermalization time and hot carrier temperature, the NLA response can be modulated flexibly. Our finding suggests that quasi-2D perovskites are a potential system for NLO applications and provide a promising way to tune NLA response of nonlinear material.

## Methods

### Materials
Lead iodide ($PbI_2$), formamidinium iodide (FAI), methylamine iodide (MAI) and phenylethylammonium iodide (PEAI) were purchased from Xi'an Polymer Light Technology Corp. Dimethyl sulfoxide (DMSO) and toluene were purchased from Sigma Aldrich. All materials were used as received.

### Perovskite film deposition
Quasi-2D perovskite precursor solution was prepared by dissolving 138.3 mg $PbI_2$, 34.4 mg FAI, and 49.82 mg PEAI (molar ratio = 3:2:2) in 1 ml DMSO. 3D perovskite precursor solution is prepared by dissolving 138.3 mg $PbI_2$, 51.6 mg FAI and 4.74 mg MAI (molar ratios = 1:0.9:0.1) in 1 ml DMSO. Small amount of MA is used to stabilize the lattice structure of 3D perovskite. The precursor solutions were stirred at room temperature for 1 h and then spin coated onto quartz substrates at 3000 rpm for 120 s. For 3D and quasi-2D perovskite films, 150 μL toluene was dropped at 30 s during spinning. Then, all films were annealed at 80 °C for 5 min to finish the deposition. Notably, to obtain high quality $(PEA)_2FAPb_2I_7$ ($n = 2$) film, FAI component ratio is slightly overweight to suppress the formation of monolayer 2D perovskite phase ($n = 1$). Minor amount of high-dimensional phase ($n > 2$) is unavoidably introduced (Supplementary Fig. 17a). However, the corresponding NLO response is found to be trivial (Supplementary Fig. 17b). Hence, the contribution from high-dimensional doping phase is negligible.

### Material characterization
The SEM images of perovskite films were measured using a Field-Emission Scanning Electron Microscope (Zigma FESEM, Zeiss, Germany) under 5 kV. AFM images were tested with the Atomic Force Microscope (Dimension Fastscan, Bruker, Germany). XRD spectra were obtained with an X-Ray Diffractometer (SmartLab 9 kW upgrade, Rigaku, Japan). The thickness of the obtained films was recorded by a Surface Profilometer (DektakXT, Bruker, Germany).

### Linear and nonlinear optical measurements
The absorption spectra of perovskite films were characterized by an Ultraviolet-vis Spectrophotometer (Perkinelmer Lambda 365). The Z-scan and transient absorption (TA) spectra were performed with

femtosecond pulse laser. The femtosecond laser source was a Coherent Astrella−1K-F Ultrafast Ti:Sapphire Amplifier (100 fs, 1 kHz, 800 nm) seeded by a Coherent Vitesse oscillator. The 540 nm-590 nm laser used in Z-scan was also obtained from the same OPA configuration. The laser pulse profile in time and frequency domain is illustrated in Supplementary Fig. 18. The Z-scan configuration is illustrated in Supplementary Fig. 19, where the sample is fixed on a one-dimensional moving stage (Zolix, KA200) and the incident laser is focused by the lens along the direction of the stags. The incident laser was split into two beams and collected by two detectors (Ophir, PD10C) to enhance the signal to noise ratio. The femtosecond TA spectra of the perovskite films were taken using the Ultrafast System HELIOS TA spectrometer. Except the 400 nm laser generating from a 0.5 mm thick BBO single crystal, the pump laser at other wavelength was also generated by the OPA system. The broadband probe pulses were generated by focusing a small portion (around 10 mJ) of the fundamental 800 nm laser pulses into a 2 mm sapphire plate.

## Data availability

The experimental data that support the findings of this study are available from the corresponding author on request.

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

## Acknowledgements

G.W. and T.L. contributed equally to this work. G.X. acknowledges the Science and Technology Development Fund, Macao SAR (File no. FDCT-0044/2020/A1, 0082/2021/A2), UM's research fund (File no. MYRG2020-00151-IAPME), the Natural Science Foundation of China (61935017, 62175268), Natural Science Foundation of Guangdong Province, China (2019A1515012186), Guangdong-Hong Kong-Macao Joint Laboratory of Optoelectronic and Magnetic Functional Materials (2019B121205002), and Shenzhen-Hong Kong-Macao Science and Technology Innovation Project (Category C) (SGDX2020110309360100). M.L. acknowledges the financial support from the Hong Kong Polytechnic University (BE2Z, W188, CD5C and ZVGH), the Shenzhen Science, Technology and Innovation Commission (R2021A064) and Research Grant Council of Hong Kong (25301522). T.L. acknowledges the start-up grant from the Department of Physics, Hong Kong Baptist University.

## Author contributions

G.X., M.L. and J.H. supervised the project. G.W. measured and analyzed the ultrafast spectroscopy. T.L. prepared the perovskite samples and performed material characterizations. B.W. and Q.W. helped with the optical tests. H.G. and Z.Z. helped with polishing the manuscript. All authors agreed with the final version of the manuscript.

## Competing interests

The authors declare no competing interests.
