## [Peer review file · Nature Communications]

REVIEWER COMMENTS

Reviewer #1 (Remarks to the Author):

Wang et al. observed the unusual power-dependent nonlinear optical absorption (NLA) conversion process from RSA (reverse-saturable absorption) to SA (saturable absorption) in quasi-2D perovskite films. Comprehensive transient absorption (TA) spectroscopy and theoretical simulation studies have been carried out to figure out the mechanism behind this abnormal phenomenon. It has been clarified that, compared to the 3D counterpart, the hot-carrier effect and accelerated hot-carrier cooling in quasi-2D perovskite play the vital role in tuning the NLA process. The work is interesting and important in terms of NLO properties and applications of low dimensional perovskites. However, some revisions need to be carried out to improve the quality of the work for publication in Nature Communications.

1. I find the preparation and characterization of the quasi-2D perovskite films not detailed and convincing enough. The authors think the prepared quasi-2D perovskite is $(\text{PEA})_2\text{FAPb}_2\text{I}_5$, by referring to Huang's work in 2016 (Nat. Photonics 10, 699-704, Ref. 25). However, this is not convincing at all, because first of all, the materials system of their work is different from Huang's in terms of the large amine (one is NMA and the other is PEA). How do they identify the formula as $(\text{PEA})_2\text{FAPb}_2\text{I}_5$, which does not even follow the general formula $\text{L}_2\text{Sn-1MnX}_{3n+1}$ as mentioned in the introduction?

It is not precise to mention the film as a monocomponent compound, as the quasi-2D perovskite films prepared by spin-coating are often multi-component, maybe with a major component as defined by the feed ratio of the starting compounds.

I strongly recommend the authors provide more preparation details and comprehensive/convincing characterizations about the quasi-2D perovskite films.

In this context, Figure 1 should be divided into two, one for characterization data of the films, and the other for the abnormal NLO characteristics (maybe with proposed mechanism as shown in Supplementary Figure 1).

2. What is the main reason arousing the accelerated hot-carrier cooling in quasi-2D perovskite, and how would the authors think the n value affect the dynamics of the hot-carriers?

3. Following the same line, would such an abnormal NLO effect applicable to similar quasi-2D perovskites, by replacing the halogen I with Br, for example? I think, maybe not as comprehensive TA studies, but at least similar Z-scan measurements should be carried out on counterpart quasi-2D perovskites.

Reviewer #2 (Remarks to the Author):

This manuscript analyzes the mechanism involving the nonlinear absorption in 2D perovskites ((PEA)₂FAPb₂I₅) polycrystalline thin films. Their most important finding is the unexpected evolution from Reverse Saturation Absorption (RSA) to Saturable Absorption (SA) with the excitation fluence, which, according to the authors, is an unexpected and firstly reported behavior in semiconductors. These results are initially studied by Z-scan and then corroborated by Transient Absorption Spectroscopy, which is also used to explain the influence of excitons and carriers in the nonlinear absorption. Finally, authors perform a theoretical model to reproduce their experimental data and to deeply understand the physics involving the experiments. All work is carried out by comparing the same experiment/theory with a bulk (PEA)₂FAPb₂I₅ semiconductor, which shows the opposite behavior and allows the authors to conclude that carrier cooling is responsible for this nonlinear absorption process. The paper is well organized, methods are well-explained, figures are clear, and conclusions well extracted. However, the manuscript also presents some drawbacks that prevent the publication in Nature Communications, at least in its current form.

1. There are some minor issues that would help the reader to better understand the paper. For example, I miss the indication of the thickness of the films in the main manuscript or the inclusion of the absorption spectra of the bulk perovskite. Besides, I think the authors should consider the possibility to include the first figure of the Supplementary Information in the manuscript (may be in figure 1). This figure clearly explains their findings and what is shown in this paper. It would be also nice to include the list of acronyms in the Supplementary Information
2. Following the first comment, I found the Supplementary Information a bit messy (if one wants to read together with the manuscript it is necessary to jump between the different sections). I think it is better to order the sections according to main manuscript. This helps the reader to read both documents at the same time.
3. The nonlinear absorption is studied here from 540 to 580 nm where the exciton resonance is located. At these conditions, Two Photon Absorption should be inhibited, and the material presents linear absorption (quite high by looking table 3). Can authors discuss if the linear absorption could prevent a practical application or not? Why RSA is only observed at short wavelengths? Do the authors expect the same behavior if the material is pumped below the bandgap (i.e. in the infrared)?
4. I do not understand the RSA coefficient of 12.65 GW/cm² claimed in page 6, are authors talking about beta coefficient or saturable intensity? Where is this coefficient obtained (I cannot find in the curves or tables) ? I do not understand the units also.
5. Can the authors add a reference to support this sentence: where the value of α is 1/3 for perovskites?
6. Figure 2c and table 2 shows negative and positive values of the beta factor. Why? I would expect a positive beta (to make the dip in the curve or RSA) and a positive I_{sat} (to promote the maximum or SA). I think it is necessary to explain how the modelling of the Z-scan curve is performed. Is this fitting

included in the manuscript? It is not clear if solid lines in Figure 1c and supplementary Figure 4 correspond to equation (2) in the main manuscript or the standard equation (1) given in the SI.

7. Can parameters in equation (2) be related with standard nonlinear coefficients? What is the meaning of parameter I_s^2 ?

8. According to authors the main novelty of the work is the evolution from RSA to SA, because bulk perovskite and III-V semiconductors show the opposite behavior (SA to RSA). However, and as far as I understand, dimensionality is necessary to observe this abnormal nonlinear absorption. Is the transition from SA to RSA also observed with other two-dimensional semiconductors (perovskite or III-V quantum wells)? This would confirm the novelty of the work, but I think is necessary to explain why. In other words, what are the conditions to observe the RSA to SA transition?

9. I miss one or two sentences before the conclusions explaining the benefits of the RSA to SA conversion compared with the standard SA to RSA behavior.

- All reviewer comments are displayed in *italics*.
- All our responses are displayed in black.
- Sentences indicating changes to the manuscript are **blue**.

Response to the reviewers' comments

Reviewer #1

Wang et al. observed the unusual power-dependent nonlinear optical absorption (NLA) conversion process from RSA (reverse-saturable absorption) to SA (saturable absorption) in quasi-2D perovskite films. Comprehensive transient absorption (TA) spectroscopy and theoretical simulation studies have been carried out to figure out the mechanism behind this abnormal phenomenon. It has been clarified that, compared to the 3D counterpart, the hot-carrier effect and accelerated hot-carrier cooling in quasi-2D perovskite play the vital role in tuning the NLA process. The work is interesting and important in terms of NLO properties and applications of low dimensional perovskites. However, some revisions need to be carried out to improve the quality of the work for publication in Nature Communications.

Response: We are delighted to hear the reviewer's recognition on the novelty and importance of our research discovery. The reviewer's critical comments are very helpful for us to further improve the manuscript. We have carefully considered the reviewer's comments and made the necessary revision as follows:

1. I find the preparation and characterization of the quasi-2D perovskite films not detailed and convincing enough. The authors think the prepared quasi-2D perovskite is (PEA)₂FAPb₂I₅, by referring to Huang's work in 2016 (Nat. Photonics 10, 699-704, Ref. 25). However, this is not convincing at all, because, first of all, the materials system of their work is different from Huang's in terms of the large amine (one is NMA and the other is PEA). How do they identify the formula as (PEA)₂FAPb₂I₅, which does not even follow the general formula L₂Sn-1MnX_{3n+1} as mentioned in the introduction? It is not precise to mention the film as a monocomponent compound, as the quasi-2D perovskite films prepared by spin-coating are often multi-component, maybe with a major component as defined by the feed ratio of the starting compounds. I strongly recommend the authors provide more preparation details and comprehensive/convincing characterizations about the quasi-2D perovskite films. In this context, Figure 1 should be divided into two, one for characterization data of the films, and the other for the abnormal NLO characteristics (maybe with proposed mechanism as shown in Supplementary Figure 1).

Response:

We thank the reviewer for pointing out our mistakes and negligence. After checking the

manuscript and supplementary information, we correct the “(PEA)₂FAPb₂I₅” to “(PEA)₂FAPb₂I₇”.

The spin-coating method is indeed difficult to prepare pure phase quasi-2D perovskite film with monocomponent compound. Nevertheless, provided adopt appropriate synthesis condition and ingredient ratio, the quasi-2D bilayer perovskite phase can be promoted to be the dominant component (*Nat. Photonics* **2016**, 10, 699-704. And *Nat. Nanotechnol.* **2016**, 11, 872-877.). Herein, the quasi-2D perovskite precursor PbI₂, FAI, and PEAI molar ratio is chosen as 3:2:2. The PEAI component ratio is slightly overweight to suppress the formation of monolayer 2D perovskite phase (n=1), because the absorption peak of n=1 2D phase is close to that of n=2 quasi-2D phase, which is possible to affect the nonlinear optical property characterization. Minor amount of high-dimensional phase (n>2) is unavoidably introduced. In the revised manuscript, we add a new figure to clarify this point. According to **Supplementary Fig. 17**, we can find the TA signal originated from high-dimensional phases is much weaker. Moreover, we also compared the TA spectra collected under 540nm and 650 nm laser pumping. The photon energy at 650 nm is smaller than the energy gap of the n=2 phase. Therefore, only the high-dimensional doping phases are excited. Here, we find that the sample only exhibits a weak bleaching with intensity around 0.2%. Hence, the influence to the NLO observation from high-dimensional doping phase should be negligible.

Supplementary Figure 17. TA spectrum of the high-dimensional doping phase. (a) Broad band TA spectra of the (PEA)₂FAPb₂I₇ quasi-2D perovskite film. The pump laser is 400 nm with intensity of 30 μJ cm⁻². Red frame highlights the weak signal from high dimensional doping phase indicating a low content. (b) TA spectra collected under 540 nm and 650 nm laser pumping at 100 fs delay. The pump intensity is maintained at ~0.4 mJ cm⁻². The photon energy of 650 nm is lower than the bandgap of bilayer quasi-2D perovskite phase. Hence, only the high-dimensional doping phase is excited. Blue circle denotes the nonlinear response of the high dimensional doping phase, which only result in a very weak bleaching with intensity around 0.2%.

We have added more preparation details in “Method” section of revised manuscript. “Notably, to obtain high quality (PEA)₂FAPb₂I₇ (n=2) film, PEAI component ratio is slightly overweight to suppress the formation of monolayer 2D perovskite phase (n=1). Minor amount of high-dimensional phase (n>2) is unavoidably introduced

(Supplementary Fig. 17a). However, the corresponding NLO response is found to be trivial (Supplementary Fig. 17b). Hence, the contribution from high-dimensional doping phase is negligible.”

A new reference is added:

27 Yuan, M. *et al.* Perovskite energy funnels for efficient light-emitting diodes. *Nat. Nanotechnol.* **11**, 872-877 (2016).

Moreover, the **Supplementary Figure 1**, **Supplementary Figure 2** in supplementary information and **Figure 1** in main text are replaced by new **Figure 1** and **Figure 2** in the revised version to demonstrated the NLA mechanisms and physical characterization. Accordingly, the main text about this section is also amended.

Figure 1. Nonlinear absorption switching mechanism. (a) Schematics of NLA switching in conventional inorganic semiconductors (top) and the corresponding NLA evolution as a function of pump intensity (bottom), according to standard saturable

absorption (SA) and reverse-saturable absorption (RSA), where the RSA can stem from two-photon absorption (2PA) and results in a conversion from SA to RSA. CB and VB denote conduction band and valence band respectively. $\Delta\alpha$ denotes the NLA induced absorption coefficient difference. (b) NLA of organic molecules based on excited state absorption (ESA) leading to an inverse transition from RSA to SA. S_{0-2} represent the separated energy level. σ is the absorption cross-section. (c) NLA process of quasi-2D (PEA)₂FAPb₂I₇ perovskite film in our work, where the competition between non-thermalized carrier induced bleaching and many-body effect induced excitonic absorption enhancement results in a conversion from RSA to SA. (d) Schematic of the initial carrier evolution process in deposited quasi-2D perovskite film under femtosecond laser pulse excitation, where the blue blocks represent the bilayer perovskite grains. The left inserted illustration demonstrates the 2D lattice structure, and the right inset illustrates the initially ultrafast intra-band relaxation of photon generated carriers. Red arrow indicates the transition from VB to CB. Solid and dash lines represent the non-thermalized carrier and hot-carrier distribution, respectively. N_{NTC} and N_{HC} denote the non-thermalized carrier and hot carrier qualities at the resonance with incident photon energy ($\hbar\omega$). T_C is hot carrier temperature reflecting the carrier distribution after thermalization. Compared with the high T_C distribution (green), low T_C distribution (orange) will reduce the value of N_{HC} and weaken the bleaching induced by band filling.

Figure 2. Physical properties and nonlinear optical characteristics. (a, b) Scanning electron microscope (SEM) and atomic force microscope (AFM) images of bilayer quasi-2D (a) and 3D (b) perovskite thin films. (c) and (d) are the corresponding X-ray diffraction (XRD) patterns and film step profiles. (e) Linear ultraviolet-vis absorbance (Abs.) spectra of the 3D perovskite and quasi-2D perovskite film with a strong 1S exciton absorption peak at 570 nm. (f) Z-scan profile (hollow circles) evolutions of bilayer quasi-2D perovskite film collected at the blue side (540 nm) of 1S exciton peak demonstrating a conversion from RSA to SA with excitation intensity rising. The solid lines are fitting curves using the abnormal NLA conversion model (Eq. 2) we developed. (g) Corresponding Z-scan results (hollow circles) of 3D counterpart at 700 nm above band-edge with a pure SA response fitting well with the standard SA model (solid lines) according to Eq. 4 in supplementary information.

2. What is the main reason arousing the accelerated hot-carrier cooling in quasi-2D perovskite, and how would the authors think the n value affect the dynamics of the hot-carriers?

Response:

According to previous research (*ACS Nano* **2019**, 13, 12621-12629), for low-dimensional quantum-wells, the accelerated hot carrier cooling was attributed to reduced coulomb screening due to dielectric confinement effect. Meanwhile, with the quantum-wells thickness decreasing, the lattice of the quasi-2D perovskite will become softer (*ACS Appl. Mater. Interfaces* **2020**, 12, 17881-17892), which may also enhance the electron-photon interaction, resulting in faster hot-carrier cooling. However, addressing specific mechanism of the ultrafast cooling observed in quasi-2D perovskite is still unambiguous. As we discussed in our manuscript, the nonequilibrium carriers could have lost most of their energy during thermalization process (within the first 100 fs). Therefore, such accelerated hot-carrier cooling in quasi-2D perovskite maybe have more complicated mechanisms which deserve further exploration. In fact, we have made some preliminary results to reveal the hot carrier effect in a series of quasi-2D perovskite with different quantum-well thickness. Here, the MA and BA are chosen as the small and large organic cation to synthesize the quasi-2D perovskite single crystal. The obtained single crystals are illustrated in **Fig. R1**. **Fig. R2a~c** are the TA spectra for the exfoliated sample. Notably, with the n value increasing, the hot carrier induced high-energy bleaching tail become much more remarkable. From **Fig. R2d**, the hot carrier tail induced bleach peak broadening can be observed more clearly in higher-dimensional quasi-2D perovskite, which indicates the stronger hot carrier effect. Besides, the suspended band-edge bleaching building up kinetics curves (**Fig. R2e**) also reflects that the hot carrier cooling become slower with quantum-well thickness increasing. This result confirms that the dimensionality may have a great influence on the hot carrier cooling process. Nevertheless, the underlying accelerated hot-carrier cooling mechanism would not affect our present work's conclusion of abnormal nonlinear absorption in quasi-2D perovskite arising from hot-carrier occupation, which is further confirmed in our Br sample as shown below in answer to Q3. We expect to uncover the detail underlying mechanism in our next stage study.

Figure R1. Optical image of the synthesized $(\text{BA})_2\text{MAPb}_2\text{I}_7$ ($n=2$), $(\text{BA})_2\text{MA}_2\text{Pb}_3\text{I}_{10}$

($n=3$) and $(\text{BA})_2\text{MA}_3\text{Pb}_4\text{I}_{13}$ ($n=4$) quasi-2D perovskite single crystal.

Figure R2. (a~c) Pseudocolour representation of TA spectra of $(\text{BA})_2\text{MAPb}_2\text{I}_7$ ($n=2$), $(\text{BA})_2\text{MA}_2\text{Pb}_3\text{I}_{10}$ ($n=3$) and $(\text{BA})_2\text{MA}_3\text{Pb}_4\text{I}_{13}$ ($n=4$) single crystals. The extra-energy of the photon-injected carrier is maintained at 0.9 eV. The pump intensity is $120 \mu\text{J cm}^{-2}$. (d) The normalized TA spectra at 500 fs probe time delay. With the n value increasing, the bleaching peak broaden remarkably to the high-energy side. (e) The normalized band-edge bleaching building up kinetics curves within the first 3 ps.

3. Following the same line, would such an abnormal NLO effect applicable to similar quasi-2D perovskites, by replacing the halogen I with Br, for example? I think, maybe not as comprehensive TA studies, but at least similar Z-scan measurements should be carried out on counterpart quasi-2D perovskites.

Response:

We thank the reviewer for this constructive comment. Here, we synthesize the $(\text{PEA})_2\text{FAPb}_2\text{Br}_7$ ($n=2$) quasi-2D perovskite film with the same spin coating method. The absorption spectrum and thickness characterization are illustrated in **Fig. R3**. A systematically Z-scan measurement is performed from the excitonic peak (435 nm) to the high-energy side (410 nm). As shown in **Supplementary Fig. 13**, the sample also exhibits apparent RSA-SA switching at the regime of 410 nm \sim 420 nm, which is in line with the phenomenon observed in $(\text{PEA})_2\text{FAPb}_2\text{I}_7$ film. It worth noting that, compared with that of $(\text{PEA})_2\text{FAPb}_2\text{I}_7$ film, the NLA conversion threshold is increased to $\sim 10 \text{ GW cm}^{-2}$. We also extracted the NLA coefficients of the sample at 410 nm and 415 nm by fitting the Z-scan curves at low pump fluence. The obtained values are 10.10 cm MW^{-1} and 23.80 cm MW^{-1} respectively, which are comparable with that of

(PEA)₂FAPb₂I₇ film at band edge (12.75 cm MW⁻¹ at 540 nm and 21.80 cm MW⁻¹ at 550 nm). This result indicates that the SA part from continuous states bleaching become much weaker in (PEA)₂FAPb₂Br₇ perovskite. We also perform TA measurement using 410 nm pump laser. As shown in **Supplementary Fig. 14**, the sample also suggests obvious absorption singularity owing to non-thermalized carriers, yet, compared with that of (PEA)₂FAPb₂I₇, the signal intensity is much weaker. This difference could be attributed to the lower continuous state quantity in the Br perovskite. Less continuous states are not enough to support strong bleaching induced by the non-thermalized carrier occupation. Hence, the SA will be weakened.

Besides the (PEA)₂FAPb₂Br₇ (n=2) quasi-2D perovskite film, we also study the NLA response of (PEA)₂PbI₄ (n=1) 2D perovskite. As illustrated in **Supplementary Fig. 12**, in (PEA)₂PbBr₄ 2D perovskite, the absorption of continuous states is further reduced. Therefore, as illustrated in **Supplementary Fig. 15**, at the blue side of the exciton peak (485 nm~495 nm), the NLA conversion phenomenon cannot be observed. The sample only suggests RSA response even though the pump power has surpassed 10 GW cm⁻². This result is also confirmed by the TA spectrum (**Supplementary Fig. 16**), where the non-thermalized carrier induced singularity almost disappeared. These results indicate that the RSA-SA conversion is not a universal property of every 2D materials. By modulating the component and dimensionality, the NLA response can be tuned flexibly for 2D perovskite materials.

Figure R3. (a) Linear ultraviolet-vis absorbance (Abs.) spectrum of (PEA)₂FAPb₂Br₇ (n=2) quasi-2D perovskite film with a strong 1S exciton absorption peak at 435 nm. (d) Corresponding step profile of the prepared quasi-2D perovskite film suggesting the thickness of the film is 68±5 nm.

The new **Supplementary Figs. 12~16** are added in supplementary information:

Supplementary Figure 12. Linear absorption spectra of different 2D perovskite materials. Absorption spectra of $(\text{PEA})_2\text{FAPb}_2\text{Br}_7$ ($n=2$), $(\text{PEA})_2\text{PbI}_4$ ($n=1$) and $(\text{PEA})_2\text{FAPb}_2\text{I}_7$ ($n=2$) perovskite films normalized at the exciton peak. The absorption from doping states and scattering have been deducted. Dash lines denote the corresponding continuous state absorption shoulder adjacent to the excitonic absorption peak.

Supplementary Figure 13. Z-scan of $(\text{PEA})_2\text{FAPb}_2\text{Br}_7$ ($n=2$) quasi-2D perovskite film. Z-scan profile evolutions of $(\text{PEA})_2\text{FAPb}_2\text{Br}_7$ ($n=2$) quasi-2D perovskite film. At the blue side (410 nm~420 nm) of 1S exciton peak (435 nm), the sample demonstrates

a conversion from RSA to SA with excitation intensity rising.

Supplementary Figure 14. TA spectrum of (PEA)₂FAPb₂Br₇ (n=2) quasi-2D perovskite film. (a) Pseudocolour representation of TA spectrum of (PEA)₂FAPb₂Br₇ (n=2) quasi-2D perovskite film pumped by 410 nm femtosecond laser with an excitation intensity of 0.45 mJ cm⁻². Inset: enlarged TA spectrum of the part labeled by the red frame. (b) TA spectra during (150 fs) and after (500 fs) thermalization.

Supplementary Figure 15. Z-scan of (PEA)₂PbI₄ (n=1) 2D perovskite film. Z-scan profile evolutions of (PEA)₂PbI₄ (n=1) 2D perovskite film at the blue side (485 nm~495 nm) of 1S exciton peak (515 nm).

Supplementary Figure 16. TA spectrum of (PEA)₂PbI₄ (n=1) 2D perovskite film.

(a) Pseudocolour representation of TA spectrum of (PEA)₂PbI₄ (n=1) 2D perovskite film pumped by 490 nm femtosecond laser with an excitation intensity of 0.38 mJ cm⁻². Inset: enlarged TA spectrum of the part labeled by the red frame. (b) TA spectra during (150fs) and after (1 ps) thermalization.

At the end of the “Discussion”, we add a new paragraph:

“To verify the universality of the phenomenon we observed, the NLA behavior of other 2D perovskite materials are also studied. We synthesize the (PEA)₂FAPb₂Br₇ (n=2) quasi-2D perovskite (replacing the iodine (I) element with bromine (Br) element) and monolayer (PEA)₂PbI₄ (n=1) 2D perovskite with similar spin-coating method. As shown in supplementary Fig. 12, the obtained (PEA)₂FAPb₂Br₇ and (PEA)₂PbI₄ film suggest remarkable 1S exciton peak at 435 nm and 515 nm. For (PEA)₂FAPb₂Br₇ perovskite, the Z-scan curves (supplementary Fig. 13) suggest the similar RSA-SA conversion at the blue side of the band-edge (410 nm ~ 420 nm), which is in line with that we observed in (PEA)₂FAPb₂I₇ film. From the TA spectrum (supplementary Fig. 14), we can also observe the ultrafast absorption singularity signal. Moreover, we also investigate the NLA properties of (PEA)₂PbI₄ (n=1) 2D perovskite film (supplementary Fig. 15). By contrast, as illustrated in supplementary Fig. 15, the sample only suggest RSA at the blue side of the exciton peak (515 nm) until high pump intensity up to 10 GW cm⁻². By retrospecting the linear absorption spectra of different 2D perovskite (supplementary Fig. 12), we find the absorption shoulder from continuous state is much lower for (PEA)₂PbI₄ (n=1) perovskite, which indicates a lower continuous state density. Low continuous state density will suppress the non-thermalized carrier induced bleaching and make the RSA to SA switching difficult to occur. This deduction is also confirmed by the TA spectrum (supplementary Fig. 16), where the transient absorption singularity signal owing to the non-thermalized carrier occupation become almost absent. These results indicate that the RSA-SA conversion is not a universal property for every 2D materials. Therefore, to realize RSA-SA conversion, besides the ultrafast carrier cooling, appropriate continuous state density is needed. By modulating the component and dimensionality, the NLA response can be tuned flexibly for 2D perovskite materials. We foresee that the abnormal RSA-SA conversion observed in quasi-2D perovskite can complement with the typical SA-RSA conversion observed in conversional semiconductors. Compared the typical SA-RSA conversion, the RSA-SA conversion is more appropriate for the application in laser pulse compressing and eliminating the low-intensity wing signal.⁵² Moreover, such abnormal NLO response is also expected to inspire novel strategy to design all-optical devices e.g. optical witching or logic-gate.”

Reviewer #2

This manuscript analyzes the mechanism involving the nonlinear absorption in 2D perovskites ((PEA)2FAPb2I5) polycrystalline thin films. Their most important finding is the unexpected evolution from Reverse Saturation Absorption (RSA) to Saturable Absorption (SA) with the excitation fluence, which, according to the authors, is an unexpected and firstly reported behavior in semiconductors. These results are initially studied by Z-scan and then corroborated by Transient Absorption Spectroscopy, which is also used to explain the influence of excitons and carriers in the nonlinear absorption. Finally, authors perform a theoretical model to reproduce their experimental data and to deeply understand the physics involving the experiments. All work is carried out by comparing the same experiment/theory with a bulk (PEA)2FAPb2I5 semiconductor, which shows the opposite behavior and allows the authors to conclude that carrier cooling is responsible for this nonlinear absorption process. The paper is well organized, methods are well-explained, figures are clear, and conclusions well extracted. However, the manuscript also presents some drawbacks that prevent the publication in Nature Communications, at least in its current form.

Response:

We are grateful to hear the reviewer's recognition on the novelty and importance of our research discovery. The reviewer's critical comments are very helpful for us to further improve the manuscript. We have carefully considered the reviewer's comments and made the necessary revision as follows:

1. There are some minor issues that would help the reader to better understand the paper. For example, I miss the indication of the thickness of the films in the main manuscript or the inclusion of the absorption spectra of the bulk perovskite. Besides, I think the authors should consider the possibility to include the first figure of the Supplementary Information in the manuscript (may be in figure 1). This figure clearly explains their findings and what is shown is this paper. It would be also nice to include the list of acronyms in the Supplementary Information.

Response:

We thank the reviewer for this constructive comment. In the revised version, the **Supplementary Figure 1, Supplementary Figure 2** in supplementary information and Figure 1 in main text are reintegrated to be new **Figure 1 and Figure 2** to demonstrate the underlying mechanisms and physical characterization. Accordingly, the main text about this section is also revised.

Figure 1. Nonlinear absorption switching mechanism. (a) Schematics of NLA switching in conventional inorganic semiconductors (top) and the corresponding NLA evolution as a function of pump intensity (bottom), according to standard SA and RSA, where the RSA can stem from two-photon absorption (2PA). CB and VB denote conduction band and valence band respectively. $\Delta\alpha$ denotes the NLA induced absorption coefficient difference. (b) NLA of organic molecules based on excited state absorption (ESA) leading to an inverse transition from RSA to SA. S_{0-2} represent the separated energy level. σ is the absorption cross-section. (c) NLA process of quasi-2D (PEA)₂FAPb₂I₇ perovskite film in our work, where the competition between non-thermalized carrier induced bleaching and many-body effect induced excitonic absorption enhancement results in a conversion from RSA to SA. (d) Schematic of the initial carrier evolution process in deposited quasi-2D perovskite film under femtosecond laser pulse excitation, where the blue blocks represent the bilayer perovskite grains. The left inserted illustration demonstrates the 2D lattice structure, and the right inset illustrates the initially ultrafast intra-band relaxation of photon

generated carriers. Red arrow indicates the transition from VB to CB. Solid and dash lines represent the non-thermalized carrier and hot-carrier distribution, respectively. N_{NTC} and N_{HC} denote the non-thermalized carrier and hot carrier qualities at the resonance with incident photon energy ($\hbar\omega$). T_C is hot carrier temperature reflecting the carrier distribution after thermalization. Compared with the high T_C distribution (green), low T_C distribution (orange) will reduce the value of N_{HC} and weaken the bleaching induced by band filling.

Figure 2. Physical properties and nonlinear optical characteristics. (a, b) Scanning electron microscope (SEM) and atomic force microscope (AFM) images of bilayer quasi-2D (a) and 3D (b) perovskite thin films. (c) and (d) are the corresponding X-ray diffraction (XRD) patterns and film step profiles. (e) Linear ultraviolet-vis absorbance (Abs.) spectra of the 3D perovskite and quasi-2D perovskite film with a strong 1S exciton absorption peak at 570 nm. (f) Z-scan profile (hollow circles) evolutions of bilayer quasi-2D perovskite film collected at the blue side (540 nm) of 1S exciton peak demonstrating a conversion from RSA to SA with excitation intensity rising. The solid

lines are fitting curves using the abnormal NLA conversion model (Eq. 2) we developed. (g) Corresponding Z-scan results (hollow circles) of 3D counterpart at 700 nm above band-edge with a pure SA response fitting well with the standard SA model (solid lines) according to Eq. 4 in supplementary information.

Besides, we add a list of acronyms in supplementary information to help reader understand our research more conveniently:

List of Acronyms

Acronyms	Definition
AFM	Atomic Force Microscope
BGR	Band-Gap Renormalization
CB	Conduction Band
HC	Hot Carrier
ESA	Excited State Absorption
MPA	Multiphoton Absorption
NLA	Nonlinear Absorption
NLO	Nonlinear Optics/Optical
NTC	Non-Thermalized Carrier
PIB	Photon-Induced Bleaching
PIA	Photon-Induced Absorption
QW	Quantum-Well
RSA	Reverse Saturable Absorption
SA	Saturable Absorption
SEM	Scanning Electron Microscope
2D	Two-Dimensional
3D	Three-Dimensional
TA	Transient Absorption
2PA	Two-Photon Absorption
VB	Valence Band
XRD	X-ray Diffraction

2. Following the first comment, I found the Supplementary Information a bit messy (if one wants to read together with the manuscript it is necessary to jump between the different sections). I think it is better to order the sections according to main manuscript. This helps the reader to read both documents at the same time.

Response:

We would like to thank the reviewer for the constructive comments. In the revised manuscript, supplementary information is ordered according to the main text. At the beginning of supplementary information, we add a catalogue for easy understanding:

Catalogue

Supplementary Note 1: Nonlinear absorption conversion mechanism.

Supplementary Table 1. Summary of the calculated RSA-SA conversion threshold of reported organic molecules under 100 fs laser pulse excitation.

Supplementary Note 2: Standard nonlinear absorption and saturable absorption fitting model.

Supplementary Figure 1. Z-scan of $(\text{PEA})_2\text{FAPb}_2\text{I}_7$ ($n=2$) quasi-2D perovskite film.

Supplementary Figure 2. Z-scan of $\text{FA}_{0.9}\text{MA}_{0.1}\text{PbI}_3$ 3D perovskite film.

Supplementary Table 2. Summary of the NLO parameters of $(\text{PEA})_2\text{FAPb}_2\text{I}_7$ quasi-2D perovskite film.

Supplementary Note 3: Initial photon-injecting carrier density calculation.

Supplementary Figure 3. Wavelength-dependent TA spectra.

Supplementary Figure 4. Power-dependent TA spectra.

Supplementary Figure 5. Wavelength-dependent absorption singularity.

Supplementary Figure 6. Power-dependent absorption singularity.

Supplementary Figure 7. Ultrafast thermalization and hot carrier cooling kinetics.

Supplementary Table 3. Summary of the fitting parameters using NLA conversion model.

Supplementary Note 4: Linear and transient absorption spectrum of semiconductor.

Supplementary Figure 8. Linear absorption spectra fitting.

Supplementary Figure 9. TA spectra fitting.

Supplementary Figure 10. TA kinetics at band-edge.

Supplementary Note 5: NLA simulation on the basis of non-thermalized and hot carrier model.

Supplementary Table 4. The parameters using in NLA simulation.

Supplementary Figure 11. NLA simulation with different parameters.

Supplementary Figure 12. Linear absorption spectra of different 2D perovskite materials.

Supplementary Figure 13. Z-scan of $(\text{PEA})_2\text{FAPb}_2\text{Br}_7$ ($n=2$) quasi-2D perovskite film.

Supplementary Figure 14. TA spectrum of $(\text{PEA})_2\text{FAPb}_2\text{Br}_7$ ($n=2$) quasi-2D perovskite film.

Supplementary Figure 15. Z-scan of $(\text{PEA})_2\text{PbI}_4$ ($n=1$) 2D perovskite film.

Supplementary Figure 16. TA spectrum of $(\text{PEA})_2\text{PbI}_4$ ($n=1$) 2D perovskite film.

Supplementary Figure 17. TA spectrum of the high-dimensional doping phase.

Supplementary Figure 18. Autocorrelation signal (a) and spectrum (b) of the used femtosecond laser pulse in experiment.

Supplementary Figure 19. The schematic diagram of the Z-scan system.

3. The nonlinear absorption is studied here from 540 to 580 nm where the exciton resonance is located. At these conditions, Two Photon Absorption should be inhibited, and the material presents linear absorption (quite high by looking table 3). Can authors discuss if the linear absorption could prevent a practical application or not? Why RSA is only observed at short wavelengths? Do the authors expect the same behavior if the material is pumped below the bandgap (i.e. in the infrared)?

Response:

We would like to thank the reviewer for the constructive comments. This is a quit significant question. Even though the linear absorption coefficient of quasi-2D perovskite is high, considering that the thickness of the film is only ~80 nm, the transmittance at 540 nm is still 45%. In another hand, the abnormal RSA-SA conversion also requires some linear absorption, because that, at the blue side of the exciton peak, the linear absorption stems from the continuous states. The SA in this regime depends on the bleaching induced by the non-thermalized carrier occupation of the continuous states. If the linear absorption is reduced sharply, the remained continuous states is not enough to support strong SA to realize the RSA-SA switching. For instance, **Supplementary Fig. 12** illustrates the linear absorption of $(\text{PEA})_2\text{PbI}_4$ ($n=1$) 2D perovskite we synthesized with spin-coating method. Compared with $(\text{PEA})_2\text{FAPb}_2\text{I}_7$ ($n=2$) quasi-2D perovskite, its exciton absorption peak become stronger, yet the continuous absorption part decreases significantly. Therefore, as illustrated in **Supplementary Fig. 15**, at the blue side of the exciton peak (485 nm~495 nm), the NLA conversion phenomenon is absent. The sample only suggest RSA response even though the pump power has surpassed 10 GW cm^{-2} . This result is also confirmed by the TA spectrum (**Supplementary Fig. 16**), where the non-thermalized carrier induced singularity almost disappear. Apparently, to obtain RSA-SA conversion in perovskite quantum-well materials, some linear absorption is required. The more detailed discussion about the influence of component and dimensionality will be demonstrated in followed answer to Q8.

In supplementary information, the **Supplementary Figs. 12, 15, 16** are added:

Supplementary Figure 12. Linear absorption spectra of different 2D perovskite materials. Absorption spectra of $(\text{PEA})_2\text{FAPb}_2\text{Br}_7$ ($n=2$), $(\text{PEA})_2\text{PbI}_4$ ($n=1$) and $(\text{PEA})_2\text{FAPb}_2\text{I}_7$ ($n=2$) perovskite film normalized at the exciton peak. The absorption from doping states and scattering have been deducted. Dash lines denote the corresponding continuous state absorption shoulder adjacent to the excitonic absorption peak.

Supplementary Figure 15. Z-scan of $(\text{PEA})_2\text{PbI}_4$ ($n=1$) 2D perovskite film. Z-scan profile evolutions of $(\text{PEA})_2\text{PbI}_4$ ($n=1$) 2D perovskite film at the blue side (485 nm~495 nm) of 1S exciton peak (515 nm).

Supplementary Figure 16. TA spectrum of $(\text{PEA})_2\text{PbI}_4$ ($n=1$) 2D perovskite film. (a) Pseudocolour representation of TA spectrum of $(\text{PEA})_2\text{PbI}_4$ ($n=1$) 2D perovskite film pumped by 490 nm femtosecond laser with an excitation intensity of 0.38 mJ cm^{-2} . Inset: enlarged TA spectrum of the part labeled by the red frame. (b) TA spectra during (150fs) and after (1 ps) thermalization.

For another question “Why RSA is only observed at short wavelengths?”. As we discussed in main text, the RSA at short wavelengths is attributed to the excitonic absorption enhancement induced by many-body effect, which only occurs above the band-edge. As shown in Fig. 3b in main text (revised version), excitonic absorption enhancement will result in a PIA_1 shoulder above the band-edge indicating a broadband RSA. Notably, at the longer wavelength below the band-edge, there are another PIA_2 signal. However, the origin of PIA_2 is different from that of PIA_1 . It can be ascribed to biexciton interaction. With reducing the pump photon energy, the amplitude of PIA_2 peak below the band edge decreases rapidly (**Supplementary Fig. 3**). Therefore, when the excitation position is below band edge, we only can observe SA at 580 nm and 590 nm rather than RSA in Z-scan measurement (**Supplementary Fig. 1**). At the regime below band-edge, the remaining possible origin of RSA is two-photon absorption.

Nevertheless, the two-photon absorption coefficient of conventional semiconductors is relatively low (normally on the order of GW cm^{-2}). Considering the thickness of the sample is below 100 nm, we think the influence from two-photon absorption is trivial. Here, we also perform a TA measurement to confirm this point. As shown in **Fig. R1**, we use 650 nm laser to pump the quasi-2D perovskite film. The photon energy of 650 nm photon is far below the band-edge of the quasi-2D bilayer perovskite phase. Here, no two-photon absorption induced ultrafast PIA signal is observed. The sample only suggests a weak PIB response with intensity around 0.2%, which can be attributed to the minor amount of remaining high-dimensional doping phase ($n > 2$).

Figure R1. (a) Pseudocolour representation of TA spectrum of $(\text{PEA})_2\text{PbI}_4$ ($n=1$) 2D perovskite film pumped by 650 nm femtosecond laser with an excitation intensity of $\sim 0.4 \text{ mJ cm}^{-2}$. (b) TA kinetics curve at 650 nm. (c) TA spectra at different delay time. Red frame denotes the pump laser excitation regime. The sample suggests negative TA signal indicating a PIB response.

4. I do not understand the RSA coefficient of 12.65 GW/cm^2 claimed in page 6, are authors talking about beta coefficient or saturable intensity? Where is this coefficient obtained (I cannot find in the curves or tables) ? I do not understand the units also.

Response:

We thank the reviewer for pointing out our mistake. We check the main text and supplementary information and revise the “ 12.65 MW cm^{-2} ” to “ 12.75 cm MW^{-1} ”, which consists with the data in Table 2.

5. Can the authors add a reference to support this sentence: where the value of α is 1/3 for perovskites?

Response:

We thank the reviewer for pointing out our oversight. We supplementary the references in main text “where the value of α is 1/3 for perovskites.^{38, 39}”

Two new references are added:

- 38 Yang, Y. *et al.* Observation of a hot-phonon bottleneck in lead-iodide perovskites. *Nat. Photonics* **10**, 53-59 (2015).
- 39 Mondal, A. *et al.* Ultrafast exciton many-body interactions and hot-phonon bottleneck in colloidal cesium lead halide perovskite nanocrystals. *Phys. Rev. B* **98**, 115418 (2018).

6. Figure 2c and table 2 shows negative and positive values of the beta factor. Why? I would expect a positive beta (to make the dip in the curve or RSA) and a positive I_{sat} (to promote the maximum or SA). I think is necessary to explain how the modelling of the Z-scan curve is performed. Is this fitting included in the manuscript? It is not clear if solid lines in Figure 1c and supplementary Figure 4 correspond to equation (2) in the main manuscript or the standard equation (1) given in the SI.

Response:

In this work, the experimental Z-scan curves of quasi-2D perovskite is fitted using three different model including NLA conversion model, standard SA model and standard NLA model:

In Fig. 3c (revised version) of main text and table 2 of supplementary information, the NLA coefficients are obtained by fitting the Z-scan data with a standard NLA model as Eq. 4 in **Supplementary Note 2**. The overall absorption coefficient of material is given by:

$$\alpha = \alpha_0 + \beta I \quad (\text{R1})$$

where α_0 is the linear absorption coefficient and β is the NLA coefficient, I is the incident laser power. Normally, Eq. R1 is used to fit RSA process to extract the RSA coefficient, and the opposite SA is described using a SA model:

$$\alpha = \frac{\alpha_s}{\left(1 + I/I_s\right)} + \alpha_u \quad (\text{R2})$$

Here, α_s is the saturable absorption component, α_u is unsaturable absorption component, I_s is the saturable absorption intensity defined as the optical intensity when the optical absorbance is reduced to half of its original value. It is worth noting that, at moderate light intensity ($I \ll I_s$), Eq. R2 can be approximate as Eq. R1:

$$\begin{aligned} \alpha &= \frac{\alpha_s}{\left(1 + I/I_s\right)} + \alpha_u \\ &= \alpha_s + \alpha_u - \frac{\alpha_s/I_s}{\left(1 + I/I_s\right)} \times I \end{aligned} \quad (\text{R3})$$

In this equation, first and second term can be regards as the linear absorption coefficient under low light field intensity as $\alpha_s + \alpha_u = \alpha_0$. When the light intensity I is far below the saturable intensity I_s , the denominator of the third terms $1 + I/I_s$ can be approximate as 1. Therefore, the SA model at moderate light intensity ($I \ll I_s$) also can be treated using Eq. R1. The NLA absorption coefficient β equals to $-\alpha_s/I_s$. Obviously,

for SA, the value of NLA absorption coefficient β is negative. Therefore, we can use a unified NLA model (Eq. R1) to study the NLA behavior changing with wavelength and avoid the influence of other NLO effects which appear under high pump intensity. As shown in **Supplementary Fig. 1** (revised version), the red dash lines denote the fitting

result using the unified NLA model (Eq. R1). The extracted NLA coefficient is exhibited in Fig. 3c of main text and table 2 of supplementary information.

In **Supplementary Fig. 1**, besides the aforesaid standard NLA model at low pump fluence, the abnormal NLA conversion model (eq. 2 in main text) and standard SA model (Eq. 6 of supplementary information) are also used to fit the Z-scan experimental data. The solid lines represent the fitting results. At the wavelength form 540 nm to 555 nm, the Z-scan curves suggest apparent RSA-SA conversion. Hence, the NLA conversion model consist well with the experimental data. However, at the wavelength above 560 nm, the RSA almost disappear and the Z-scan curves suggest a pure SA behavior. Therefore, at this regime 560 nm~590 nm, the Z-scan data is fitted with a standard SA model. The extracted SA parameters such as saturable intensity and modulation depth are also listed in table 2. Likewise, the fitting lines of Fig. 2f and g in main text also use two different models. Fig. 2f is fitted according to the abnormal NLA conversion model, and Fig. 2g is fitted using the standard SA model.

In the revised manuscript, we make a more detailed explanation to avoid misunderstanding:

In section of **Supplementary Note 2**. The first sentence is amended by:

“For a typical reverse saturable absorption (RSA), the overall absorption coefficient can be expressed as:

$$\alpha = \alpha_0 + \beta I \quad (4)$$

Where α_0 is the linear absorption coefficient and β is the NLA coefficient, I is the incident laser power. The OA Z-scan curves could be fitted by:⁸

$$T_{\text{norm}} = \frac{1}{\sqrt{\pi}[\beta I(z)L_{\text{eff}}]} \int_{-\infty}^{+\infty} \ln[1 + \beta I(z)L_{\text{eff}}\exp(-t^2)] dt \quad (5)$$

”

At the end of this section, we add a new sentence:

“It is worth noting that, at moderate light intensity ($I \ll I_s$), Eq. 6 can be approximate as Eq. 4, where the linear absorption coefficient under low light field intensity as $\alpha_0 = \alpha_s + \alpha_u$, and the NLA absorption coefficient β equals to $-\alpha_s/I_s$. Obviously, for SA, the value of NLA absorption coefficient β is negative. Therefore, under moderate light intensity ($I \ll I_s$), Eq. 4 can also be regards as a standard NLA model applying to both RSA and SA process.”

Furthermore, we provide more explanation of **Supplementary Fig. 1** in the revised figure caption:

“Supplementary Figure 1. Z-scan of (PEA)₂FAPb₂I₇ (n=2) quasi-2D perovskite film. OA Z-scan measurement of (PEA)₂FAPb₂I₇ quasi-2D perovskite film at 540 nm ~ 590 nm under different excitation intensity. The peak power at focus (0 position) of each curve is shown in top right corner. Hollow circles represent the experimental data. Solid lines are the theoretical fitting curves. Here, two different models are adapted. For 540 nm ~ 555 nm, the Z-scan results suggest apparent RSA-SA switching. Therefore, during this regime, the experimental data is fitted according to the abnormal nonlinear conversion model as Eq. 2 in main text. At wavelength above 560 nm, the

RSA part is so weak that can be neglected. Hence, we adapt a standard saturable absorption model (Eq. 6). Additionally, we also adapt a standard nonlinear absorption model (Eq. 4) to extract the NLA coefficient at different wavelength under moderated pump fluence. The red dash lines denote the fitting curves. The details can be found in **Supplementary Note 2**. The calculated parameters including nonlinear absorption coefficient, saturable intensity and modulation depth are exhibited in **Supplementary Table 2**.”

And **Fig. 2** caption in the revised main text:

“(f) Z-scan profile (hollow circles) evolutions of bilayer quasi-2D perovskite film collected at the blue side (540 nm) of 1S exciton peak demonstrating a conversion from RSA to SA with excitation intensity rising. The solid lines are fitting curves using the abnormal NLA conversion model (Eq. 2) we developed. (g) Corresponding Z-scan results (hollow circles) of 3D counterpart at 700 nm above band-edge with a pure SA response fitting well with the standard SA model (solid lines) according to Eq. 4 in supplementary information.”

7. Can parameters in equation (2) be related with standard nonlinear coefficients? What is the meaning of parameter I_{s2} ?

Response:

Under particular condition, Eq. 2 can be simplified as the standard NLA model. As we discussed in manuscript, the abnormal RSA-SA conversion can be described with the abnormal NLA conversion model (Eq. 2 in main text). The RSA and SA part can be tuned by the saturable intensity I_{s1} and I_{s2} , respectively. Hence, we can discuss in two different situations. Firstly, at low pump intensity meeting $I < I_{s1} < I_{s2}$. Both the denominators of first and second term can be treated as 1 and Eq. 2 can be approximately expressed as the standard NLA model:

$$\alpha = (\beta - a_0/I_{s2}) \times I \tag{R4}$$

The $(\beta - a_0/I_{s2})$ can be regarded as an effective NLA coefficient β_{eff} .

Moreover, when the light intensity meets $I_{s1} < I < I_{s2}$. The denominators in first term cannot be omitted. The Eq. 2 can be approximated as the Eq. 3 in main text, which is the situation we encounter in this work. If we compare Eq.3 in main text and standard SA-RSA conversion model (Eq. 2 in supplementary information), we can find these two models have the same form, yet the position of the denominator, which induce the saturation effect, changes. In **Fig. R2**, we demonstrate such difference induced changing of the NLA evolution.

Figure R2. Demonstration between the abnormal RSA-SA conversion model and conventional SA-RSA model. With the denominator position changing, the NLA conversion behavior is reversed.

In main text, we introduce the parameter I_{s2} into the second term of equation (2). That is because, as the TA measurement result, the SA under high pump intensity is attributed to the non-thermalized carrier occupation induced ultrafast bleaching. Thus, it still adapts to the SA model to describe it. The I_{s2} can be defined as the value of pump intensity to provide enough non-thermalized carrier occupation to induce 50% bleaching of absorption. Here, the value of I_{s2} reached up to 40.64 GW cm⁻² because the ultrafast thermalization by carrier-carrier interaction will force the injected non-thermalized carriers to move away from their initial energy position. The non-thermalized carrier is difficult to cumulate. Therefore, higher pump intensity is required to realize SA, which leads to the high value of saturable intensity I_{s2} .

8. According to authors the main novelty of the work is the evolution from RSA to SA, because bulk perovskite and III-V semiconductors show the opposite behavior (SA to RSA). However, and as far as I understand, dimensionality is necessary to observe this abnormal nonlinear absorption. Is the transition from SA to RSA also observed with other two-dimensional semiconductors (perovskite or III-V quantum wells)? This would confirm the novelty of the work, but I think is necessary to explain why. In other words, what are the conditions to observe the RSA to SA transition?

Response:

Besides the (PEA)₂FAPb₂I₇ (n=2) quasi-2D perovskite, we also studied nonlinear behavior of other 2D perovskite materials as shown in the answer to Q3 of reviewer 2. The NLA of (PEA)₂PbI₄ (n=1) 2D perovskite film demonstrates pure RSA response. Moreover, we replace the I element with Br element to synthesized the (PEA)₂FAPb₂Br₇ (n=2) quasi-2D perovskite film (**Fig. R3**). The Z-scan results of (PEA)₂FAPb₂Br₇ film suggest similar RSA-SA conversion with pump intensity increasing, as shown in **supplementary Fig. 13**. From the TA spectrum (**supplementary Fig. 14**), we can find the sample also illustrate the nonlinear absorption singularity signal.

Figure R3. (a) Linear ultraviolet-vis absorbance (Abs.) spectrum of $(\text{PEA})_2\text{FAPb}_2\text{Br}_7$ ($n=2$) quasi-2D perovskite film with a strong 1S exciton absorption peak at 435 nm. (d) Corresponding step profile of the prepared quasi-2D perovskite film suggesting the thickness of the film is 68 ± 5 nm.

We also try to investigate the NLA properties of high-dimensional perovskite ($n>2$) with larger quantum well thickness. However, obtaining pure phase high-dimensional perovskite with spin-coating method is challenging.

Furthermore, we note that such RSA-SA switching is rare for other conventional 2D materials such as graphene and transition metal sulfide. For instance, the graphene is reported as a broadband saturable absorber (*ACS nano*, **2012**, 6, 3677-3694), and the hot carrier effect is strong in graphene (*Physical review letters*, **2009**, 102, 086809). The initial hot carrier temperature even surpasses 2000 K, which supports the reported broadband SA of graphene materials. Similarly, other 2D materials such as MoS_2 , black phosphorus *et. al.* are also reported to demonstrate broadband SA (*Adv. Mater.* **2018**, 30, 1705963).

Thus, the observation of RSA-SA conversion in our quasi-2D perovskite is novel. The specific NLA response is influenced by many factors. To realize RSA-SA conversion, the materials should possess ultrafast hot carrier cooling to avoid the contribution of initial high-temperature hot carrier induced bleaching. Meanwhile, the density of state adjacent to band edge should not be too low. Otherwise, the SA will become too weak to switch over the NLA. By modulating the structure and component, we can design specific materials with desired NLO properties.

At the end of “Discussion” section, a new paragraph is added:

“To verify the universality of the phenomenon we observed, the NLA behavior of other 2D perovskite materials are also studied. We synthesize the $(\text{PEA})_2\text{FAPb}_2\text{Br}_7$ ($n=2$) quasi-2D perovskite (replacing the iodine (I) element with bromine (Br) element) and monolayer $(\text{PEA})_2\text{PbI}_4$ ($n=1$) 2D perovskite with similar spin-coating method. As shown in supplementary Fig. 12, the obtained $(\text{PEA})_2\text{FAPb}_2\text{Br}_7$ and $(\text{PEA})_2\text{PbI}_4$ film suggest remarkable 1S exciton peak at 435 nm and 515 nm. For $(\text{PEA})_2\text{FAPb}_2\text{Br}_7$ perovskite, the Z-scan curves (supplementary Fig. 13) suggest the similar RSA-SA conversion at the blue side of the band-edge (410 nm ~ 420 nm), which is in line with that we observed in $(\text{PEA})_2\text{FAPb}_2\text{I}_7$ film. From the TA spectrum (supplementary Fig.

14), we can also observe the ultrafast absorption singularity signal. Moreover, we also investigate the NLA properties of $(\text{PEA})_2\text{PbI}_4$ ($n=1$) 2D perovskite film (supplementary Fig. 15). By contrast, as illustrated in supplementary Fig. 15, the sample only suggest RSA at the blue side of the exciton peak (515 nm) until high pump intensity up to 10 GW cm^{-2} . By retrospecting the linear absorption spectra of different 2D perovskite (supplementary Fig. 12), we find the absorption shoulder from continuous state is much lower for $(\text{PEA})_2\text{PbI}_4$ ($n=1$) perovskite, which indicates a lower continuous state density. Low continuous state density will suppress the non-thermalized carrier induced bleaching and make the RSA to SA switching difficult to occur. This deduction is also confirmed by the TA spectrum (supplementary Fig. 16), where the transient absorption singularity signal owing to the non-thermalized carrier occupation become almost absent. These results remind us that the RSA-SA conversion is not a universal property for every 2D materials. Therefore, to realize RSA-SA conversion, besides the ultrafast carrier cooling, appropriate continuous state density is needed. By modulating the component and dimensionality, the NLA response can be tuned flexibly for 2D perovskite materials.”

In the supplementary information, the **supplementary Fig. 13, 14** of new data are added:

Supplementary Figure 13. Z-scan of $(\text{PEA})_2\text{FAPb}_2\text{Br}_7$ ($n=2$) quasi-2D perovskite film. Z-scan profile evolutions of $(\text{PEA})_2\text{FAPb}_2\text{Br}_7$ ($n=2$) quasi-2D perovskite film. At the blue side (410 nm~420 nm) of 1S exciton peak (435 nm), the sample demonstrates a conversion from RSA to SA with excitation intensity rising.

Supplementary Figure 14. TA spectrum of (PEA)₂FAPb₂Br₇ (n=2) quasi-2D perovskite film. (a) Pseudocolour representation of TA spectrum of (PEA)₂FAPb₂Br₇ (n=2) quasi-2D perovskite film pumped by 410 nm femtosecond laser with an excitation intensity of 0.45 mJ cm⁻². Inset: enlarged TA spectrum of the part labeled by the red frame. (b) TA spectra during (150 fs) and after (500 fs) thermalization.

9. *I miss one or two sentences before the conclusions explaining the benefits of the RSA to SA conversion compared with the standard SA to RSA behavior.*

Response:

We think the abnormal RSA to SA conversion we observed in quasi-2D perovskite is a significant supplement for standard SA to RSA behavior. For instance, in the laser pulse shaping field, the standard SA to RSA is used to obtain the long-range square pulse. By contrast, the abnormal RSA to SA conversion could be used to pulse compression or eliminate the low-intensity wing signal. Here we add a sentence before the conclusion: “We foresee that the abnormal RSA-SA conversion observed in quasi-2D perovskite can complement with the typical SA-RSA conversion observed in conversional semiconductors. Compared the typical SA-RSA conversion, the RSA-SA conversion is more appropriate for the application in laser pulse compressing and eliminating the low-intensity wing signal.⁵¹ Moreover, such abnormal NLO response is also expected to inspire novel strategy to design all-optical devices *e.g.* all-optical switching or logic-gate.”

A new reference is added:

52 Bao, Q. & Loh, K. P. Graphene photonics, plasmonics, and broadband optoelectronic devices. *ACS Nano* **6**, 3677-3694 (2012)

REVIEWERS' COMMENTS

Reviewer #1 (Remarks to the Author):

The authors have made extensive revisions, which have addressed most of my concerns. I believe the manuscript is now ready for publication.

Reviewer #2 (Remarks to the Author):

Now the authors have addressed all the questions and I think the paper is ready for publication I specially like the schemes of Figure 1 and the arrangement of the supplementary information. I have only found the minor comments listed above:

1. Page 3. A typo in the last sentence "have be".
2. Page 7. Replace "than the previous" by "than the previous one".
3. Page 15. Replace "is collected" by "are collected".
3. Figures. The letters of some of the labels could be enlarged to allow a better reading.

- All reviewer comments are displayed in *italics*.
- All our responses are displayed in black.
- Sentences indicating changes to the manuscript are blue.

Response to the reviewers' comments

Reviewer #1

The authors have made extensive revisions, which have addressed most of my concerns. I believe the manuscript is now ready for publication.

Response: We are delighted to hear the reviewer's recognition on the novelty and importance of our research discovery. Thanks to reviewer providing critical comments to further improve our manuscript.

Reviewer #2

Now the authors have addressed all the questions and I think the paper is ready for publication I specially like the schemes of Figure 1 and the arrangement of the supplementary information. I have only found the minor comments listed above:

Response:

We are grateful to hear the reviewer's recognition on the novelty and importance of our research discovery. Thanks to reviewer providing critical comments to further improve our manuscript.

1. 1. Page 3. A typo in the last sentence "have be".

Response:

We thank the reviewer for pointing out our mistakes and negligence. This typo has been corrected:

“Such low-dimensional QWs structures with strong quantum and dielectric confinement²² have **been** demonstrated to result in enhanced NLO responses²³⁻²⁵.”

2. Page 7. Replace "than the previous" by "than the previous one".

Response:

We replace “than the previous” by “than the previous one” in this sentence:

“The obtained RSA coefficient at 540 nm is 12.75 cm MW⁻¹ which is at least two-orders of magnificent larger **than the previous one** reported two-photon absorption (2PA) of conventional semiconductors and 3D bulk perovskites”

3. Page 15. Replace "is collected" by "are collected".

Response:

We replace “is collected” by “are collected” in this sentence:

“All the parameters used **are collected** in supplementary Table 4.”

4. Figures. The letters of some of the labels could be enlarged to allow a better reading.

Response:

We update the figures in main text and supplementary information to make sure they meet the requirement of the journal guideline.